# Oncolytic viruses engineered to enforce cholesterol efflux restore tumor-associated macrophage phagocytosis and anti-tumor immunity in glioblastoma

Shiqun Wang[1,2,8], Wei Yan[3,4,8], Lingkai Kong[1,8], Shuguang Zuo[5], Jingyi Wu[1], Chunxiao Zhu[2,6], Huaping Huang[3,4], Bohao He[1], Jie Dong ®[1] ✉ & Jiwu Wei ®[1,7] ✉

The codependency of cholesterol metabolism sustains the malignant progression of glioblastoma (GBM) and effective therapeutics remain scarce. In orthotopic GBM models in male mice, we identify that codependent cholesterol metabolism in tumors induces phagocytic dysfunction in monocyte-derived tumor-associated macrophages (TAMs), resulting in disease progression. Manipulating cholesterol efflux with apolipoprotein A1 (ApoA1), a cholesterol reverse transporter, restores TAM phagocytosis and reactivates TAM-T cell antitumor immunity. Cholesterol metabolomics analysis of in vivo-sorted TAMs further reveals that ApoA1 mediates lipid-related metabolic remodeling and lowers 7-ketocholesterol levels, which directly inhibits tumor necrosis factor signaling in TAMs through mitochondrial translation inhibition. An ApoA1-armed oncolytic adenovirus is also developed, which restores antitumor immunity and elicits long-term tumor-specific immune surveillance. Our findings provide insight into the mechanisms by which cholesterol metabolism impairs antitumor immunity in GBM and offer an immunometabolic approach to target cholesterol disturbances in GBM.

Cancer cells can develop a dependence on certain enzymes or transcription factors through altered biochemical and signaling states, even if these factors are not oncogenic. This process, called non-oncogene addiction or codependence[1]. Importantly, co-dependencies can be determined by the local biochemical environment in which tumor cells grow. GBM is a highly complex malignancy and remains difficult to treat despite advances in therapy, largely due to inherent barriers associated with the brain as its host organ[2].

The brain, which contains 20% of the body's cholesterol, relies mainly on astrocyte-driven cholesterol synthesis due to the blood–brain barrier[3]. GBM cells, being metabolically dependent on exogenous cholesterol uptake, are vulnerable to cholesterol modulators that target their cholesterol intake or astrocyte-driven cholesterol efflux[4–7]. LXR-623, a liver X receptor (LXR) β agonist, selectively kills GBM cells by inhibiting low-density lipoprotein receptor (LDLR) expression to reduce cholesterol uptake and promoting ATP-binding

[1]State Key Laboratory of Pharmaceutical Biotechnology, Medical School of Nanjing University, Nanjing, Jiangsu, China. [2]Zhejiang Cancer Hospital, Hangzhou Institute of Medicine (HIM), Chinese Academy of Sciences, Hangzhou, Zhejiang, China. [3]Department of Neurosurgery, The Second Affiliated Hospital, School of Medicine, Zhejiang University, Hang Zhou, Zhejiang, China. [4]Clinical Research Center for Neurological Diseases of Zhejiang Province, Hangzhou, Zhejiang, China. [5]Liuzhou Key Laboratory of Molecular Diagnosis, Guangxi Key Laboratory of Molecular Diagnosis and Application, Affiliated Liutie Central Hospital of Guangxi Medical University, Liuzhou, Guangxi, China. [6]School of Molecular Medicine, Hangzhou Institute for Advanced Study, UCAS, Hangzhou, Zhejiang, China. [7]Jiangsu Key Laboratory of Molecular Medicine, Medical School of Nanjing University, Nanjing, Jiangsu, China. [8]These authors contributed equally: Shiqun Wang, Wei Yan, Lingkai Kong. ✉e-mail: dongjie@nju.edu.cn; wjw@nju.edu.cn

cassette subfamily A member 1 (ABCA1) expression to increase cholesterol efflux. However, its clinical translation for GBM treatment was hindered by adverse CNS events in clinical trials[6,8]. Therefore, exploiting cholesterol metabolism in GBM to achieve tumor control remains a challenge, and reoptimizing the selectivity or considering alternative strategies may improve outcomes for GBM patients.

Targeting tumor-infiltrating immune cells is an alternative strategy for GBM treatment, as evidence suggests that cholesterol homeostasis is strongly linked to immune cell function[9]. Cholesterol accumulation in infiltrating CD8+ T cells leads to exhaustion or impaired signaling, while tumor cells induce immune tolerance by depriving TAMs of cholesterol[10–12]. Given the striking feature of GBMs in cholesterol metabolism, it is reasonable to infer that the cholesterol metabolism codependency of GBM affects immune cell fate within the tumor microenvironment (TME). Therefore, pharmacological modulation of immune cell metabolism may provide targeted treatment for GBM patients, but the role of cholesterol in the TME immune cells of GBM is not yet fully understood. Developing agents that target immune metabolism dysregulation in GBM is challenging, requiring analysis of metabolic differences in various immune subgroups and potential biological effects.

In this work, we examine the alterations in cholesterol levels with tumor progression in intracranial GBM mouse models. Further investigations reveal that abnormal accumulation of cholesterol in monocyte-derived TAMs induces upregulation of ABCA1/G1 cholesterol efflux receptors and "don't eat me" receptors, which leads to TAM phagocytic fragility. We therefore exploit ectopic expression of the ABCA1/G1 ligand ApoA1 to enhance cholesterol efflux from TAMs in the GBM microenvironment, which leads to a TAM-T-cell-mediated tumor clearance. We elucidate that cholesterol efflux reduces impairment of mitochondrial translation by a toxic cholesterol metabolite, 7-ketocholesterol. Finally, we develop an oncolytic adenovirus carrying ApoA1 and evaluate its safety and efficacy in preclinical models.

## Results

### Metabolism codependency in GBM facilitates cholesterol pooling in the TME

We quantified total cholesterol levels in fresh tumor tissue and tissue interstitial fluid of three orthotopic GBM models (Fig. 1a), which included a human GBM cell model (U251-MG), a mouse GBM cell model (GL261), and a rat GBM cell model (C6), to assess cholesterol levels in the TME of GBM. As reported previously[13], isolation of tissue interstitial fluid did not lead to significant cell lysis, as evidenced by lactate dehydrogenase (LDH) activity measurements in brain, plasma, and interstitial fluid (Supplementary Fig. 1a, b). Our analysis revealed that LDH activity in brain interstitial fluid (BIF) and tumor interstitial fluid (TIF) was less than 1% of brain activity.

We observed that the total cholesterol content in GBM tumor tissues (mean of 4.97–5.69 μg/mg tissue) was higher than that of peripheral spleen tissues (mean of 0.81–1.20 μg/mg tissue), but lower than that in normal brain tissues (mean of 6.56–7.27 μg/mg tissue) (Fig. 1b). Notably, we found that cholesterol levels in the TIF (mean of 0.37–0.54 μg/mg tissue) were significantly higher than those in the BIF (mean of 0.05–0.10 μg/mg tissue) (Fig. 1c). indicating that cholesterol was redistributed and enriched in the TME.

Cholesterol can be transported out of cells by ATP-binding cassette transporters and delivered to other cells as lipoproteins in the brain[14]. Astrocytes are known to support GBM progression by supporting tumor cell metabolism-dependent cholesterol[7]. We hypothesized that the enrichment of cholesterol in the TIF could be due to active cholesterol transportation. To test this, we measured high-density lipoprotein (HDL)-like and low/very low-density lipoprotein (LDL/VLDL) cholesterol levels in TIF from three GBM models (Fig. 1d). We found that HDL-like cholesterol levels (mean of 0.29–0.39 μg/mg

tissue) were much higher than LDL/VLDL cholesterol levels (mean of 0.03–0.05 μg/mg tissue).

Apolipoprotein E (ApoE) is a key HDL-like cholesterol transporter in the brain[15]. In intracranial murine GBM model, we found ApoE accumulation in the GBM tumor tissue margin, which was significantly higher than in adjacent tissue (Fig. 1e). Consistently, ApoE levels were higher in GBM tissues compared to adjacent tissues in patient tissue chips (Fig. 1f). Moreover, analysis of genes related to cholesterol uptake and synthesis in TCGA-GBM showed lower sterol regulatory element-binding protein 2 (SREBP2) expression for cholesterol synthesis, but higher epidermal growth factor receptor (EGFR) and LDLR expression for cholesterol uptake in GBM compared to normal brain tissue (Fig. 1g). Similar results were also observed in two intracranial GBM models (Supplementary Fig. 1c). Together, our findings suggest that the GBM cells form a cholesterol pool in the TME due to metabolic codependence (Fig. 1h).

### Cholesterol in the TME induces phagocytic fragility in monocyte-derived macrophages

Disrupted cholesterol homeostasis is crucial in maintaining an immunosuppressive microenvironment in tumors[9,10]. To investigate the role of cholesterol in infiltrating immune cells, we measured cholesterol contents in cancer cells and major immune subsets in orthotopic GBM model (Fig. 2a). Cholesterol levels in macrophages, including monocyte-derived tumor-associated macrophages (MN-TAMs) and CNS-resident macrophages (microglia), were significantly higher than those in tumor-infiltrating monocytes and lymphocytes. Notably, TAMs had 3.17-fold higher cholesterol levels than tumor-infiltrating monocytes (MNs). Interestingly, we found that cholesterol levels in TAMs were 2.0–2.6-fold higher than those in peripheral macrophages isolated from the spleen (Fig. 2b), indicating that tumor-infiltrating macrophages accumulate cholesterol. We observed a shift towards peripheral MN-TAMs over central microglia in the GBM population as the disease progressed (Fig. 2c). Therefore, we focused on investigating MN-TAMs.

Metabolic status controls immune cell differentiation, polarization, mobilization and immunity[16,17]. For example, intratumoral $T_{reg}$ cells undergo lipid metabolism reprogramming to enforce functional specialization in the TME[18]. Macrophages primarily serve to eliminate foreign bodies, microorganisms, and cellular waste through phagocytosis and digestion. TAMs can exhibit phagocytic fragility, which is marked by the decline of professional phagocytic functions[19]. To investigate the impact of cholesterol accumulation on macrophages, we examined the morphology of TAMs and peripheral macrophages. Our results showed that TAMs exhibited larger or "fattier" than splenic macrophages (Fig. 2d). To reveal this, we performed transmission electron microscopy of sorted mouse and human TAMs, which showed that the cytoplasmic contents of these cells were predominantly large or foamy phagosomes (Supplementary Fig. 2a). We observed uncleared organelles or cell debris in phagosomes (Fig. 2e and Supplementary Fig. 2a). These findings indicate that cholesterol accumulation in TAMs might be either attributed to intracellular uncleared phagosomes or to the dysregulation of phagocytosis.

Considering that the phagocytic fitness of macrophages is crucial for maintaining tissue homeostasis, we investigated whether cholesterol determines the phagocytosis of macrophages. We found that TAMs exhibited a weaker phagocytic capacity for cholesterol uptake than bone marrow-derived macrophages (BMDMs) in cocultivation of nitrobenzoxadiazole (NBD)-cholesterol-labeled tumor cell debris (Fig. 2f). This indicated that phagocytosis-mediated cholesterol uptake was reduced in TAMs. Consistently, we observed that TAMs had distinct clumps of heterochromatin in their nuclei (Fig. 2e and Supplementary Fig. 2a). These regions have denser chromatin (darker) and relatively low rates of gene transcription[20]. Furthermore, excessive cholesterol filling or treatment with TIF significantly impaired the

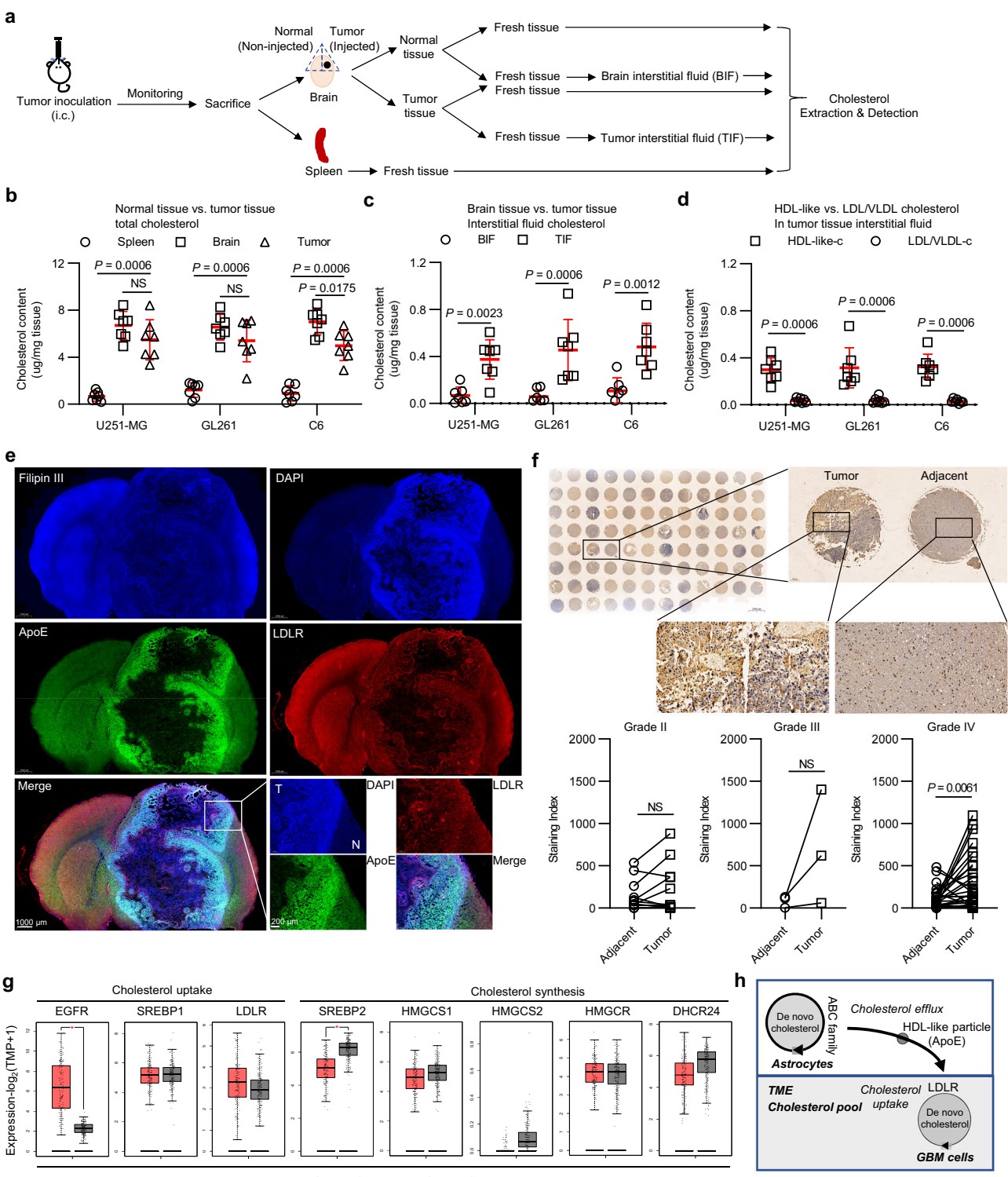

phagocytic efficacy of BMDMs (Fig. 2g, h). In line with the findings, cholesterol-saturated CD45.1 BMDMs were phagocytically incompetent after adoptively transferred into GL261^YFP-bearing CD45.2 mice (Fig. 2i, j). Taken together, these data indicate that the GBM infiltrative MN-TAMs accumulate cholesterol and become phagocytosis anergy.

## Cholesterol levels in TAMs are positively correlated with "don't eat me" receptor expression

To elucidate the mechanisms underlying the cholesterol accumulation in TAMs, we examined the expression of sterol uptake receptors,

including LDLR, CD36, and CD64, on various cell types in the TME (Supplementary Fig. 2b). We found that TAMs expressed higher levels of CD36 and CD64 compared to other cell types, while LDLR expression was comparable between TAMs and GBM cells. These results suggest that cholesterol accumulation in TAMs might be attributed to multiple cholesterol uptake pathways, including the scavenger receptor-mediated phagocytosis pathway and the LDLR-HDL-like pathway.

As phagocytosis in TAMs was inhibited despite high expression of phagocytosis-promoting receptors (CD36 and CD64), we investigated whether phagocytosis-inhibiting receptors were feedback-upregulated.

**Fig. 1 | Metabolic codependency of glioblastoma cells on cholesterol uptake facilitates the establishment of a cholesterol reservoir within the TME.** **a**–**d** Experimental setup (**a**) to study the cholesterol metabolic status in the TME of GBM models. Quantification of cholesterol levels in fresh tissues (**b**) and brain/tumor tissue interstitial fluids (**c**). Fresh spleen, brain, and tumor tissues as well as brain tissue interstitial fluid (BIF) and tumor tissue interstitial fluid (TIF) were collected and subjected to cholesterol extraction and quantification. **d** Quantification of HDL-like cholesterol levels and LDL/VLDL cholesterol levels in TIFs. $n = 7$ mice per group. **e** Representative immunofluorescence images of GBM tumors. Frozen sections from the coronal plane of C6 brain tissue were incubated with DAPI (blue), Filipin III (cholesterol dye, blue), anti-ApoE (green), and anti-LDLR (red). The images are representative of 2 tumors. **f** Representative immunohistochemistry images

and quantitative analysis of ApoE on a glioma tissue chip. The chip (bioaitech, N093Ct01) contains tumor tissues and adjacent tissues from 46 patients (grade II [$n = 12$], grade (III) [$n = 3$], grade IV [$n = 31$]). **g** GEPIA2 expression analysis of cholesterol uptake-related genes and cholesterol synthesis-related genes. Data match the TCGA-GBM and GTEx datasets. Red box indicates tumor samples; gray box indicates normal samples. Data are presented as median with interquartile range. $Log_2FC = 1$. Statistical significance was determined using one-way ANOVA with Tukey test, $P$ value = 0.05. **h** Schematic diagram of GBM cholesterol metabolism codependency. Statistical significance was determined using the Mann–Whitney test (Two-tailed) in **b**, **c**, **d**, or the paired $t$ test (Two-tailed) in **f**. Data shown are the mean ± SD. Source data are provided in the Source Data file.

We examined the expression of known phagocytosis-inhibitory receptors, such as signal regulatory protein α (SIRP-α), sialic acid binding Ig like lectin 10 (Siglec-10), and programmed cell death 1 (PD-1) in GBM tumors from two murine GBM-bearing mice (Supplementary Fig. 2c). We found that Siglec-10 and PD-1 expression in TAMs were significantly higher than that in splenic macrophages, whereas SIRP-α expression was lower in TAMs than in peripheral macrophages. Additionally, PD-1 was found to significantly impair phagocytosis of TAMs, consistent with previous studies (Supplementary Fig. 2d)[21].

We investigated whether cholesterol accumulation increases the expression of Siglec-10 or PD-1, which hampers phagocytosis of TAMs. We observed a positive correlation between the expression levels of Siglec-10 and PD-1 and cholesterol contents. The cholesterol levels of Siglec-10$^+$/PD-1$^+$ TAMs were significantly higher than those of Siglec-10$^-$/PD-1$^-$ TAMs in GBM patients and murine GBM models (Fig. 2k, l); and immunofluorescence further confirmed the colocalization of cholesterol and siglec-10 of TAMs isolated from intracranial GL261 model (Fig. 2m). To investigate the cause of increased receptor expression on TAMs, we transferred in vitro differentiated CD45.1 BMDMs into tumor-bearing CD45.2 mice to determine if infiltration into a cholesterol-rich TME was responsible. Our results showed that macrophages infiltrated into tumor tissue had increased cholesterol levels and upregulation of Siglec-10 and PD-1 expression, while those in spleen tissue did not (Fig. 2n). In vitro experiments also confirmed that cholesterol repletion increased these receptor expression levels, while β-cyclodextrin (β-CD) cholesterol depletion decreased it (Supplementary Fig. 2e, f). Overall, these findings suggest that cholesterol accumulation induces phagocytic fragility of tumor-infiltrating macrophages by upregulating "don't eat me" receptors.

## Enhancing cholesterol efflux enables tumor control in a macrophage–T-cell immune-codependent manner

Having shown that excess cholesterol leads to macrophage phagocytic fragility, we explored the possibility of reinvigorating macrophage functional specialization by restoring cholesterol homeostasis in the TME. To this end, we analyzed the gene expression of *APOA1* and *ABCA1/G1* (Fig. 3a), which are the main receptors of ApoA1 for cholesterol efflux, in GBM and normal brain tissue samples from the Gene Expression Profiling Interactive Analysis (GEPIA) database (http://gepia.cancer-pku.cn/). Our findings showed that *APOA1* expression was lower in GBM tissues compared to normal tissues, while *ABCA1/G1* expression was significantly higher than in GBM tissues, indicating a cholesterol metabolic disturbance in GBMs. Since HDL is a well-defined reverse cholesterol transporter and ApoA1 is a major component of HDL that mediates cholesterol transportation[22], our study highlights the potential of targeting the ApoA1/ABCA1/G1 pathway to restore cholesterol homeostasis in the TME and enhance macrophage function for improved tumor control.

Our previous retrospective study revealed a negative correlation between the expression level of ABCA1 on TAMs and the prognosis of wild-type isocitrate dehydrogenase (IDH$^{WT}$) GBM patients[23]. To further investigate cholesterol metabolism in GBMs, we examined the

expression patterns of APOA1 and ABCA1/G1 on GBM cells and tumor-infiltrating immune cells. Our results showed that APOA1 was almost absent in GBM cell lines (Fig. 3b), while GBM cells exhibited low expression of ABCA1/G1 in vitro and in vivo (Fig. 3c and Supplementary Fig. 3a). Similarly, peripheral immune cells in the spleen and lymph nodes also expressed low levels of ABCA1/G1 (Supplementary Fig. 3b, c). However, tumor-infiltrating immune cells, including myeloid cells and lymphocytes, showed high expression levels of ABCA1/G1 (Supplementary Fig. 3d, e), which increased ~3-fold in tumor-infiltrating lymphocytes (Supplementary Fig. 3e, f) and over ~7-fold in tumor-infiltrating monocytes compared to peripheral cells (Supplementary Fig. 3d). Notably, the expression levels of ABCA1/G1 were over ~17-fold higher in tumor-infiltrating macrophages than in peripheral macrophages, indicating that cholesterol metabolism is under upward pressure in TAMs.

We then explored the potential of enhancing cholesterol efflux by targeting ApoA1 to improve antitumor responses in GBMs. To assess whether ApoA1 directly affects GBM cell proliferation, we treated GBM cells with ApoA1 in vitro and found no significant changes in proliferation, except for a slight decrease in GL261 (Fig. 3d). To further evaluate the antitumor efficacy of ApoA1 in vivo, we established two APOA1-expressing GBM cell lines (GL261$^{APOA1}$ and G422$^{APOA1}$) and observed prolonged survival only in immunocompetent mice bearing orthotopic APOA1-expressing GBMs, but not in immunodeficient mice (Fig. 3e). Tumor control was associated with a reduced cholesterol levels in the TME, as evidenced by lower cholesterol levels in TIFs of mice bearing APOA1-expressing GBMs compared to those bearing puromycin-expressing GBMs (Fig. 3f). These findings indicate that T lymphocytes participate in ApoA1-mediated tumor control.

To confirm the role of APOA1 in manipulating antitumor immunity, we depleted immune cells in immunocompetent mice bearing orthotopic GL261$^{APOA1}$ GBM (Fig. 3g). Macrophage depletion, in addition to CD8$^+$ T-cell depletion, abolished the therapeutic benefits of ApoA1, while CD4$^+$ T or NK cell depletion had no impact on survival. We also investigated whether macrophages and CD8$^+$ T cells coexist spatially to achieve ApoA1-mediated tumor control. To this end, we cocultivated GL261$^{OVA-Luc}$ with either the activated Naive T or OT-I T cells (which exerted specific cytotoxicity on OVA-expressing GBM cells) in the presence or absence of TAMs. Indeed, treatment with ApoA1 alone did not increase tumor cell clearance in TAMs or T cells alone, but cocultivation of TAMs and T cells in the presence of ApoA1 significantly increased tumor kill (Fig. 3h). Interestingly, ApoA1 significantly reversed the immunosuppression of TAM on OT-1 cells (Fig. 3h, right panel). These results demonstrate that ApoA1-mediated tumor control targets immune cells rather than tumor cells and that macrophages and CD8$^+$ T cells interdependently participate in ApoA1-mediated antitumor responses.

## ApoA1-enforced cholesterol efflux subverts phagocytic fragility in TAMs

Having shown that ApoA1 reduced the immunosuppression of TAM on tumor-specific OT-1 cells (Fig. 3h), we next sought to delineate how ApoA1 manipulates the TAM-T-cell axis for GBM tumor control. Our

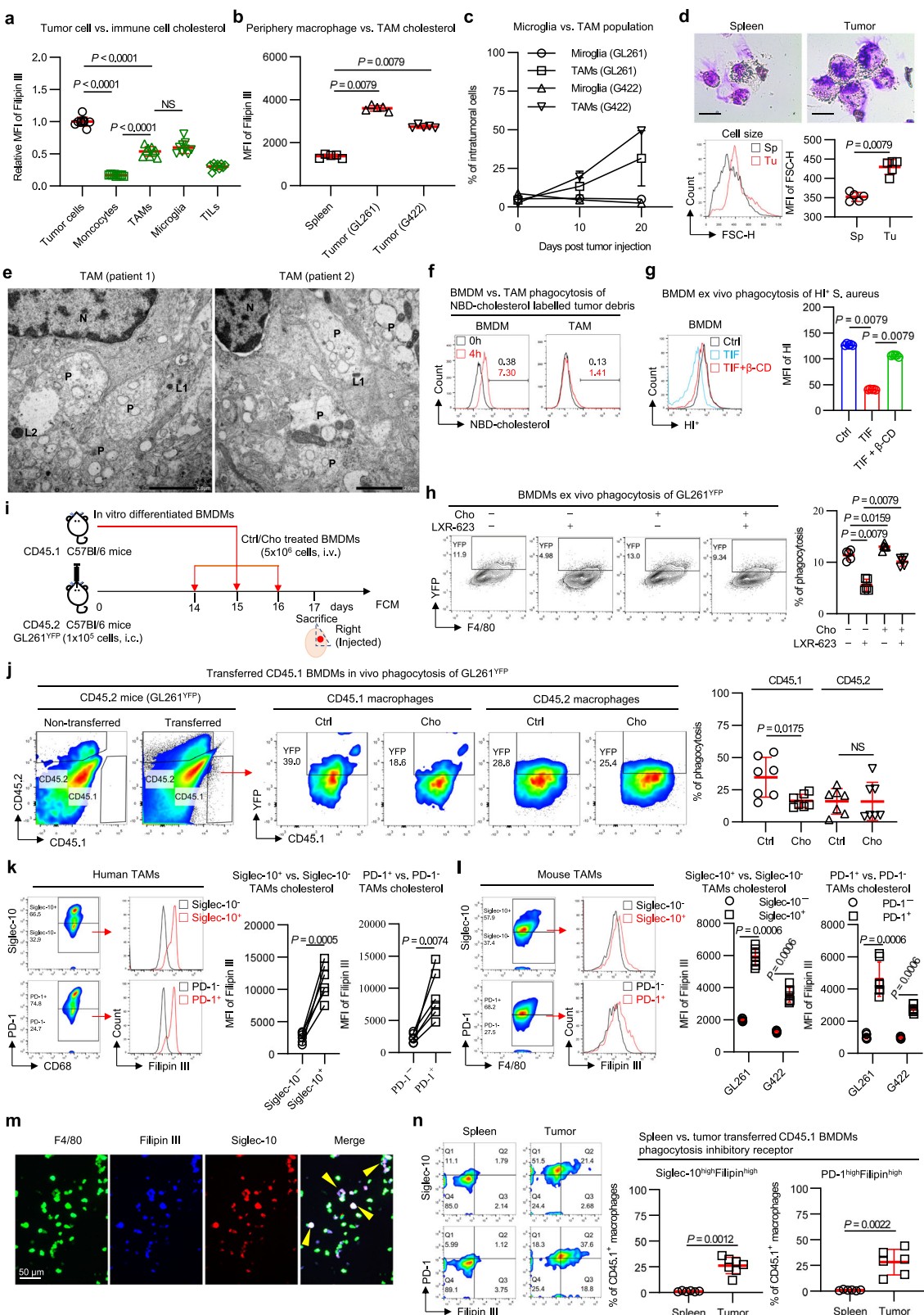

findings showed that ApoA1 was more effective in reducing cholesterol content in TAMs than in T cells (Fig. 4a), and the increased cholesterol efflux from TAMs was partially blocked by DIDS, an inhibitor of ABCA1 (Fig. 4b). These results indicate that TAMs in GBM tissues are the primary targets of ApoA1. TAMs are known to support tumor progression, but they can also exhibit intrinsic antitumor effects and promote cytotoxic T-cell activation, particularly in response to pharmacological

agents that enhance their phagocytic or oxidative functions[24,25]. We hypothesize that ApoA1 activates the TAM-T-cell axis by alleviating cholesterol-induced phagocytic fragility of TAMs.

Then we investigated the regulatory effect of ApoA1 on TAM phagocytosis in vivo and in vitro. Our results showed that ApoA1 treatment enhanced mouse and human TAM phagocytosis ex vivo (Fig. 4c, d), and this process was dependent on ApoA1-mediated

**Fig. 2 | Cholesterol accumulation induces phagocytic fragility in monocyte-derived macrophages. a** Quantification of intracellular cholesterol levels in cell subsets within intracranial GL261-bearing C57BL/6 J model. $n = 9$ mice per group. **b** Quantification of cholesterol levels in splenic macrophages and tumor macrophages within intracranial GL261-bearing C57BL/6J model and G422-bearing KM model. $n = 5$ mice per group. **c** Quantification of populations of centrally resident macrophages (microglia) and monocyte-derived TAMs during tumor progression. $n = 5$ mice per group. **d** Representative Giemsa staining and quantification of the cell size of peripheral splenic macrophages and TAMs. The upper images are representative of 2 spleen samples and 2 tumor samples. Scale bar, 20 µm. The lower FCM plots represent 5 spleen samples and 5 GL261 tumor samples. **e** Representative TEM images of human TAMs. The nuclei (N, upper left) of TAMs showed distinct heterochromatin clumps (dark); the primary (L1) and secondary (L2) lysosomes, and large, poorly cleared phagosomes (P) were observed within the cytoplasm. The images are representative of 2 samples. **f** Representative flow plots comparing phagocytosis-mediated cholesterol uptake by TAMs and BMDMs. $n = 2$ independent experiments. Quantification of ex vivo phagocytosis in TIF-treated (**g**) and cholesterol-treated (**h**) BMDMs. $n = 5$ biological samples. Experimental setup (**i**) and quantification of in vivo phagocytosis (**j**) in adoptively transferred cholesterol-treated macrophages. $n = 7$ mice per group. Quantification of cholesterol levels in CD11b[+]CD68[+] human TAMs (**k**) and CD11b[+]F4/80[+] mouse TAMs (**l**) with high or low phagocytosis-inhibitory receptors. Preparation of single-cell suspensions from patient GBM tumors ($n = 6$) and mouse GBM tumors ($n = 7$). **m** Representative immunofluorescence images of TAMs. F4/80 (green), Filipin III (blue), and Siglec-10 (red). Experiment conducted once, $n = 3$ samples. **n** Quantification of cholesterol levels and phagocytosis-inhibitory receptor expression in macrophages. Spleen, $n = 6$. Tumor, $n = 7$. Statistical significance was determined using the Mann–Whitney test (Two-tailed) in **a**, **b**, **d**, **g**, **h**, **j**, **l**, **n**, or the paired $t$ test (Two-tailed) in **k**. Data shown are the mean ± SD. Source data are provided in the Source Data file.

cholesterol efflux, as ApoA1-enhanced phagocytosis in TAMs was inhibited under ABCA1 inhibitor blockade (Fig. 4c). Phagocytosis is regulated by "don't eat me" signals such as Siglec-10 and PD-1, which are membrane-bound molecules that regulate macrophage phagocytosis[21,26]. We found that elevating local ApoA1 concentrations decreased Filipin[high]/Siglec-10[high] TAMs and Filipin[high]/PD-1[high] TAMs in orthotopic GBMs expressing APOA1 (Fig. 4e) and that TAMs exhibited significantly increased phagocytosis in these GBMs compared to control GBMs (Fig. 4f). These data demonstrate that ApoA1 enhances cholesterol efflux in TAMs and consequently restores their phagocytic function in GBMs.

## Metabolism-reinvigorated TAMs orchestrate CD8[+] T cells to clear tumor cells

To further clarify the underlying crosstalk between metabolism-reprogrammed macrophages and T cells, we examined the pro-inflammatory polarization of TAMs, including tumor necrosis factor (TNF)-α and inducible nitric oxide synthase (iNOS) production and major histocompatibility complex (MHC)-I/II expression (Fig. 4g). TNF-α production was slightly increased in GL261[APOA1] TAMs but not in G422[APOA1] TAMs, and there were no changes in iNOS production. However, MHC-I and MHC-II expression on TAMs was significantly increased in both GBMs. The loss of MHC-I antigen presentation in cancer cells contributes to immune evasion[27], while cross-presentation of antigens by neighboring macrophages is critical for activating infiltrating effector CD8[+] T cells[28–30]. We found that ApoA1 treatment promoted cell-to-cell contacts between TAMs and effector T cells, indicating the formation of tight immune synapses (Fig. 4h).

Robust phagocytosis by TAMs is critical for durable adaptive immune responses[30–33]. Thus, we investigated whether increased cell contacts between TAMs and effector T cells promoted T-cell proliferation. ApoA1 treatment did not enhance CD8[+] T-cell proliferation without cell-to-cell contacts with TAMs and tumor cells (Fig. 4i), suggesting that cytokine networks were dispensable for the TAM-T-cell axis. However, ApoA1 treatment significantly promoted CD8[+] T-cell proliferation when cells with TAMs, CD8[+] T cells, and tumor cells were in contact (Fig. 4j). Additionally, ApoA1 treatment did not improve CD8[+] T-cell proliferation in the absence of TAMs (Fig. 4k). Consistently, the population of interferon (IFN)-γ[+]/TNF-α[+] CD8[+] T cells was significantly increased in APOA1-expressing GBM tumors (Fig. 4l), while the population of PD-1[+]lymphocyte activating 3[+] (LAG-3)[+] CD8[+] T cells was significantly decreased (Fig. 4m). These data demonstrate that immune synaptic crosstalk between TAMs and CD8[+] T cells is essential for ApoA1-mediated antitumor immunity.

Ma et al. demonstrated that cholesterol present in the TME leads to exhaustion of CD8[+] T cells and impairs immune responses against tumors[10]. To investigate whether locally elevated ApoA1 directly affects TILs, we examined the effect of ApoA1 on cholesterol in tumor-infiltrating CD8[+] T cells (Fig. 4n). Together, these data indicate that ApoA1-induced changes in cholesterol metabolism rejuvenate the functional specialization of TAMs, which in turn promotes immune responses mediated by CD8[+] T cells against tumors (Fig. 4o).

## ApoA1 reprograms lipid metabolism in TAMs and prevents 7-ketocholesterol-induced mitochondrial translational inhibition

Having shown that ApoA1 targets TAMs to facilitate antitumor immune responses, we next elucidated the underlying mechanism by which ApoA1 metabolically revitalizes TAMs. Transcriptome analysis of TAMs revealed significant downregulation of lipid metabolism-related pathways in TAMs from GL261[APOA1] tumors relative to GL261[Puro] tumors (Fig. 5a, b), particularly those regulating sterol metabolism (Fig. 5c). Furthermore, cholesterol-targeted metabonomic analysis identified a significant reduction in cholesterol and oxysterol derivatives, including 7-ketochlesterol (7K-Cho) and 24(S),25-epoxycholesterol (Epoxy-Cho), in GL261[APOA1] TAMs (Fig. 5d–f). These findings demonstrate that ApoA1 reprograms TAM lipid metabolism, leading to a decrease in cholesterol and its metabolites.

We investigated whether ApoA1-enforced cholesterol efflux affects X-box binding protein 1-endoplasmic reticulum (XBP1-ER) stress levels in TAMs, given that high plasma cholesterol levels contribute to metabolic diseases by inducing ER stress[34]. However, APOA1 expression did not reduce ER stress in TAMs, as shown by XBP1 s levels (Supplementary Fig. 4a, b). Further analysis of TAM RNA-seq data revealed that mitochondrial RNA metabolism was the most significantly upregulated biological process in GL261[APOA1] TAMs compared to GL261[Puro] TAMs (Fig. 5g). In line, we examined TAMs from GL261[Puro] or GL261[APOA1] GBMs using electronic microscopy to evaluate their mitochondrial size and quantity. TAMs from GL261[APOA1] tumors exhibited larger mitochondrial size and more mitochondrial matrix compared to those from GL261[Puro] tumors (Fig. 5h), indicating that cholesterol-stress-relieved TAMs have more active mitochondrial metabolism. Mitochondria are the energy center of cells and are closely related to phagocytosis[35]. Treatment of BMDMs with cholesterol or TIF downregulated 12 of 15 genes involved in mitochondrial RNA metabolism (Fig. 5i). Furthermore, treatment with cholesterol, specifically 7K-Cho, significantly reduced mitochondrial protein content and mitochondria in BMDMs (Fig. 5j, k). These results indicate that the accumulation of cholesterol, particularly 7K-Cho, in macrophages leads to mitochondrial translation disorders.

Mitochondria contain 1000 to 1500 proteins distributed in different subcompartments[36]. We performed SDS−PAGE electrophoresis to analyze cytoplasmic and mitochondrial protein profiles in macrophages and determine which molecular weight range of protein levels in mitochondria was affected by 7K-Cho (Fig. 5l). We found that 7K-Cho treatment resulted in lower mean lane intensity for mitochondrial proteins, particularly within the 10−35 kDa molecular weight range that includes enzymes in the mitochondrial matrix and peripheral proteins attached to the mitochondrial membrane surface, such as

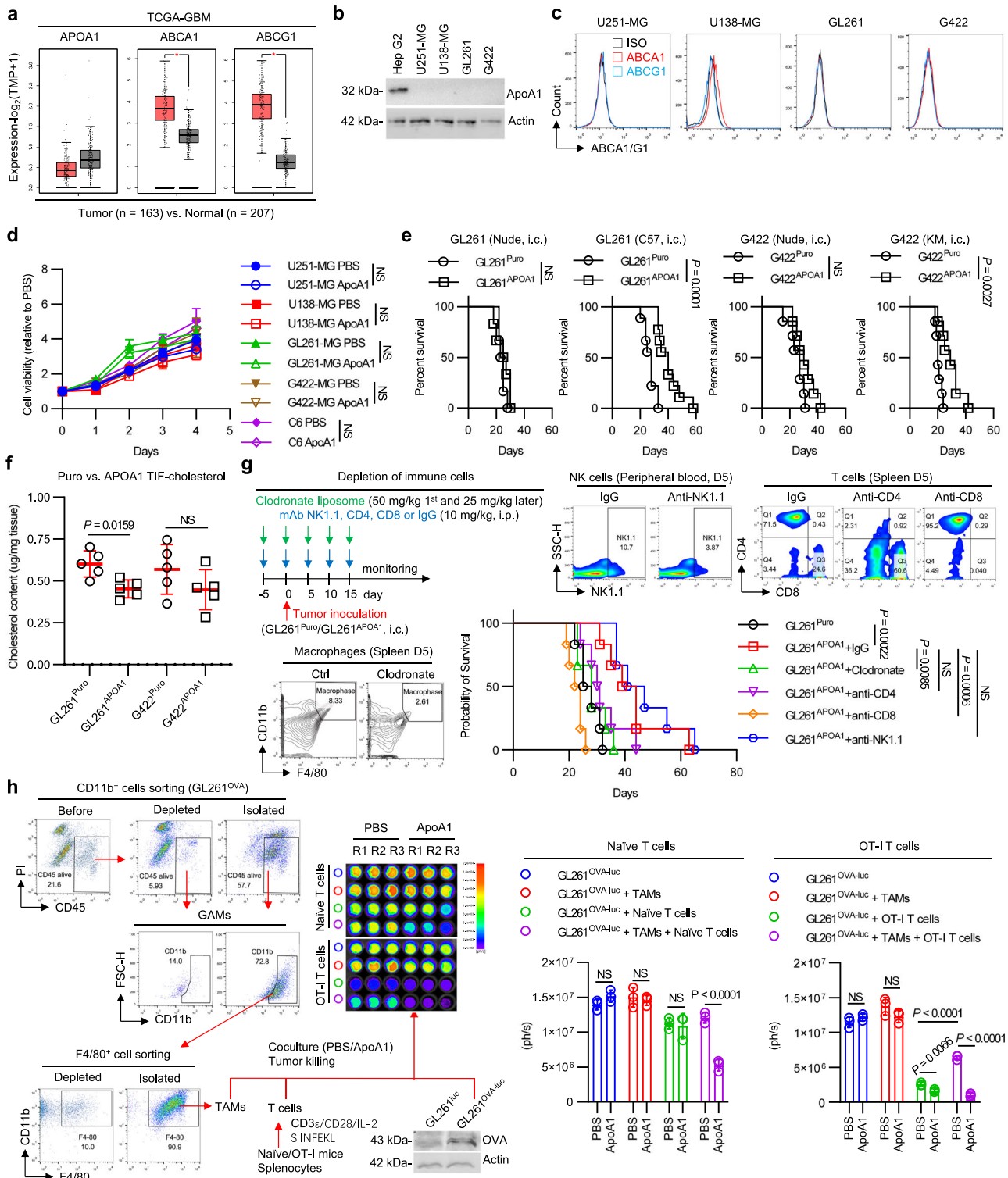

NAD(P) transhydrogenase (11 kDa) and cytochrome C (10–13 kDa)[37,38]. 7K-Cho is a toxic cholesterol oxide that can cause mitochondrial dysfunction and release free radicals[39]. We treated isolated BMDM mitochondria with 7K-Cho and observed cavitation and loss of mitochondrial matrixes (Fig. 5m), as well as a loss of membrane potential in BMDMs (Fig. 5n). Excessive cholesterol or 7K-Cho treatment of BMDMs increased mitophagy (Supplementary Fig. 4c). These findings demonstrate that 7K-Cho damages mitochondrial membranes and causes mitochondrial content spillage.

We further investigated whether ApoA1 could protect TAM mitochondria from injury by reducing cholesterol and 7K-Cho levels in

TAMs. Indeed, GL261[APOA1] TAMs had higher mitochondrial protein levels (full lane and 10–35 kDa line) than GL261[Puro] TAMs (Fig. 5o). We also examined if cholesterol or its derivatives impair macrophage phagocytic fragility by affecting mitochondrial metabolism. Mitochondrial inhibition perturbed macrophage phagocytosis, and ApoA1 reversed the impairment of phagocytosis caused by excessive cholesterol or 7K-Cho treatment (Supplementary Fig. 4d). Consistently, Treatment with an excess of 7K-Cho inhibited the recovery of ApoA1 to the phagocytosis function of TAMs (Fig. 5p). Together, these findings suggest that ApoA1-reprogrammed cholesterol metabolism protects TAM mitochondria, leading to enhanced phagocytosis.

**Fig. 3 | Elevating local ApoA1 concentrations enable tumor control in a macrophage–T-cell immune-codependent manner. a** GEPIA2 expression analysis of APOA1 and its receptors. Data match the TCGA-GBM and GTEx datasets. Data are presented as median with interquartile range. Log₂FC = 1. Statistical significance was determined using one-way ANOVA with Tukey test, $P$ value = 0.05. **b** Protein expression analysis of APOA1 in GBM cell lines cultured in vitro. $n = 2$ independent experiments. **c** Representative flow plots of APOA1 receptor ABCA1/G1 expression levels in GBM cell lines cultured in vitro. The FCM plots are representative of 3 independent experiments. **d** Cell proliferation assay for ApoA1-treated GBM cells. GBM cells ($5 \times 10^3$) were seeded in 96-well plates and cultured with either vehicle or ApoA1 (10 μg/ml) in 1% FBS medium for 0–4 days. Cell proliferation kinetics were identified by MTT assay. $n = 3$ independent experiments. **e** Kaplan–Meier survival curves of intracranial GBM^Puro- or GBM^APOA1-bearing immunodeficient and immunocompetent mice. GL261 in Nude, $n = 6$ mice per group. GL261 in C57, $n = 9$ mice per group. G422 in Nude and KM, $n = 7$ mice per group. **f** Quantification of interstitial fluid cholesterol content in APOA1-expressing intracranial GBM tumor tissues. $n = 5$ mice per group. **g** Experimental setup, validation and Kaplan–Meier survival curves of intracranial GL261^APOA1-bearing C57BL/6J mice with depletion of immune cells. $n = 6$ mice per group. **h** Determination of TAM-T-cell codependency in tumor killing. Data are from 3 independent experiments, and each point represents the mean of triplicate wells. Statistical significance was determined using the log-rank test in **e**, **g**, two-way ANOVA with Sidak's test in **d**, the Mann–Whitney test (Two-tailed) in **f**, or the unpaired $t$ test (Two-tailed) in **h**. Data shown are the mean ± SD. Source data are provided in the Source Data file.

## 7-Ketocholesterol-disturbed mitochondrial translation accelerates the disorder of TNF signaling pathway in macrophages

To uncover the mechanism of how 7K-Cho-mediated mitochondrial translation inhibition affects macrophage phagocytosis, we conducted RNA-seq analysis on in vitro differentiated BMDMs treated with 7K-Cho and doxycycline (DOX), a mitochondrial translation inhibitor (Fig. 6a, b). The results revealed that TNF signaling pathway was significantly enriched in BMDMs treated with both inhibitors. TNF-α, a pro-inflammatory activated biomarker, plays a vital role in regulating the phagocytic action of macrophages[40]. We further focused on elucidating how cholesterol or 7K-Cho affects the TNF signaling pathway in macrophages. The results showed that cholesterol and 7K-Cho inhibited the production of TNF-αand the activation of the TNF-α-induced NF-κB pathway, but not the IL-6-induced activation of the STAT3 pathway in THP-1 macrophages (Fig. 6c). Additionally, the restoration of ApoA1-mediated phagocytosis was blocked by a TNF-α inhibitor (CC5013), suggesting that ApoA1-improved phagocytosis depends on the mitochondria-TNF-signaling pathway axis (Fig. 6d).

## Oncolytic adenovirus armed with a cholesterol modulator promotes GBM clearance

Antineoplastic drugs, particularly oncolytic viruses (OVs), can induce cell death and release large amounts of cellular debris in the TME[41]. We confirmed that oncolysis of OVs led to macrophage cholesterol accumulation and phagocytic fragility (Supplementary Fig. 5a–c). TAMs are abundant in GBM, and since ApoA1-mediated cholesterol efflux mainly targets TAMs and improves antitumor immune responses, we introduced the human *APOA1* gene into a replicative type 5 adenovirus (AdV^APOA1, Fig. 7a, b). Since type 5 adenoviruses infect mammalian cells via coxsackievirus and adenovirus receptors (CAR) or integrin receptors[42], we constructed several CAR-expressing murine GBM cell lines (GL261^CAR and G422^CAR) to evaluate the therapeutic efficacy of AdV^APOA1 (Supplementary Fig. 6a, b). AdV^APOA1 selectively infected tumor cells, produced viral progeny, and secreted ApoA1 protein in orthotopic GBM tumors (Fig. 7c, d). AdV^APOA1 showed comparable levels of replication and oncolysis to AdV^Ctrl in GBM cell lines (Supplementary Fig. 6c, d), indicating that APOA1 arming did not alter viral virulence.

In several orthotopic GBM models, AdV^APOA1 intratumoral administration significantly suppressed tumor growth, prolonged survival, and maintained body weights compared to AdV^Ctrl treatment in murine GL261^CAR- and G422^CAR-bearing mice (Fig. 7e–i) as well as in immune-reconstituted humanized GBM mice (Fig. 7j–l). Of note, AdV^APOA1 achieved a complete response rate of 42% in the GL261^CAR GBM model (Fig. 7h). We further investigated the therapeutic efficacy in immune-reconstituted mice bearing human U138-MG^luc GBMs, where AdV^APOA1 effectively inhibited tumor growth and achieved a remarkable complete response rate of 71% (Fig. 7l).

## Intratumoral administration of AdV^APOA1 exhibits safety in tumor-bearing mice

Considering the adverse CNS events observed in LXR-623 clinical trials[8], we evaluated the potential toxicity associated with ApoA1 interventions. In vitro, ApoA1 did not exhibit toxicity to mouse neural stem cells (NE-4C), while the LXR agonist induced cell death significantly (Supplementary Fig. 7a). This difference may be attributed to the distinct mechanisms of cholesterol deprivation mediated by ApoA1 and LXR-623 (Supplementary Fig. 7b). In orthotopic GL261^APOA1 GBM tumors, no significant neuronal damage was detected in the para-cancerous tissues (Supplementary Fig. 7c), and no significant damage was observed in the hearts and lungs (Supplementary Fig. 7d). Furthermore, favorable bridging was observed in the AdV^APOA1-cured GBM mice (Supplementary Fig. 7e).

Moreover, in the long-term toxicity evaluation of AdV^APOA1 (Supplementary Fig. 7f), we observed that antiviral antibody titers were significantly increased after repeated dosing for 2 weeks and persisted throughout the recovery period (Supplementary Fig. 7g). Only IFN-γ cytokine secretion was slightly elevated after 2 weeks of repeated dosing, indicating that intracranial administration of AdV^APOA1 did not pose a risk of cytokine release syndrome (Supplementary Fig. 7h). This may be attributed to the fact that the virus is primarily distributed in the brain at the injection site and is shed from the feces and saliva. (Supplementary Fig. 7i). Histopathological examination at the end of the administration period (D16) revealed that viral groups exacerbated the administration-related reactions, such as cortical/white matter degeneration/necrosis, meningeal inflammatory cell infiltration, and spleen red and/or white pulp cells decreased relative to the vehicle group (Supplementary Fig. 7j). After a 4-week convalescence (D44), these side effects were significantly resolved.

Additionally, toxicity assessment experiments in rhesus monkeys by the maximum exposure route (subcutaneous and intravenous) showed that the high-dose group ($1.5 \times 10^{12}$ VP/kg, s.c.) had a severe toxicity rate of 10% (STD10), resulting in death (Supplementary Fig. 7k). The autopsy of the dead monkey revealed intermediate hepatic necrosis with inflammatory cell infiltration in the liver, cardiac papillary muscle hemorrhage with focal neutrophil infiltration, and spleen hemorrhage/congestion (Supplementary Fig. 7l). The histopathology of other virus-treated animals showed slight liver lesions, which recovered within a four-week recovery period. Together, these data demonstrate that an oncolytic adenovirus engineered to express APOA1 achieves superior tumor clearance with a laudable safety profile.

## Intratumoral administration of AdV^APOA1 reprograms TME lipid metabolism and repairs TAM phagocytosis fragility

We investigated whether AdV^APOA1 intratumoral administration could reshape the metabolic and/or immunologic microenvironment in orthotopic GBMs. Our results showed that AdV^APOA1 treatment significantly altered the expression levels of 733 genes ($P$ value = 0.05, log₂FC = 2) in GL261^CAR tumor bulks compared to AdV^Ctrl treatment (Fig. 7m). KEGG enrichment analysis of these differentially expressed genes revealed that AdV^APOA1 treatment downregulated the lipid metabolism pathway relative to AdV^Ctrl treatment (Fig. 7n), suggesting that the intratumoral administration of AdV^APOA1 reprograms TME lipid metabolism.

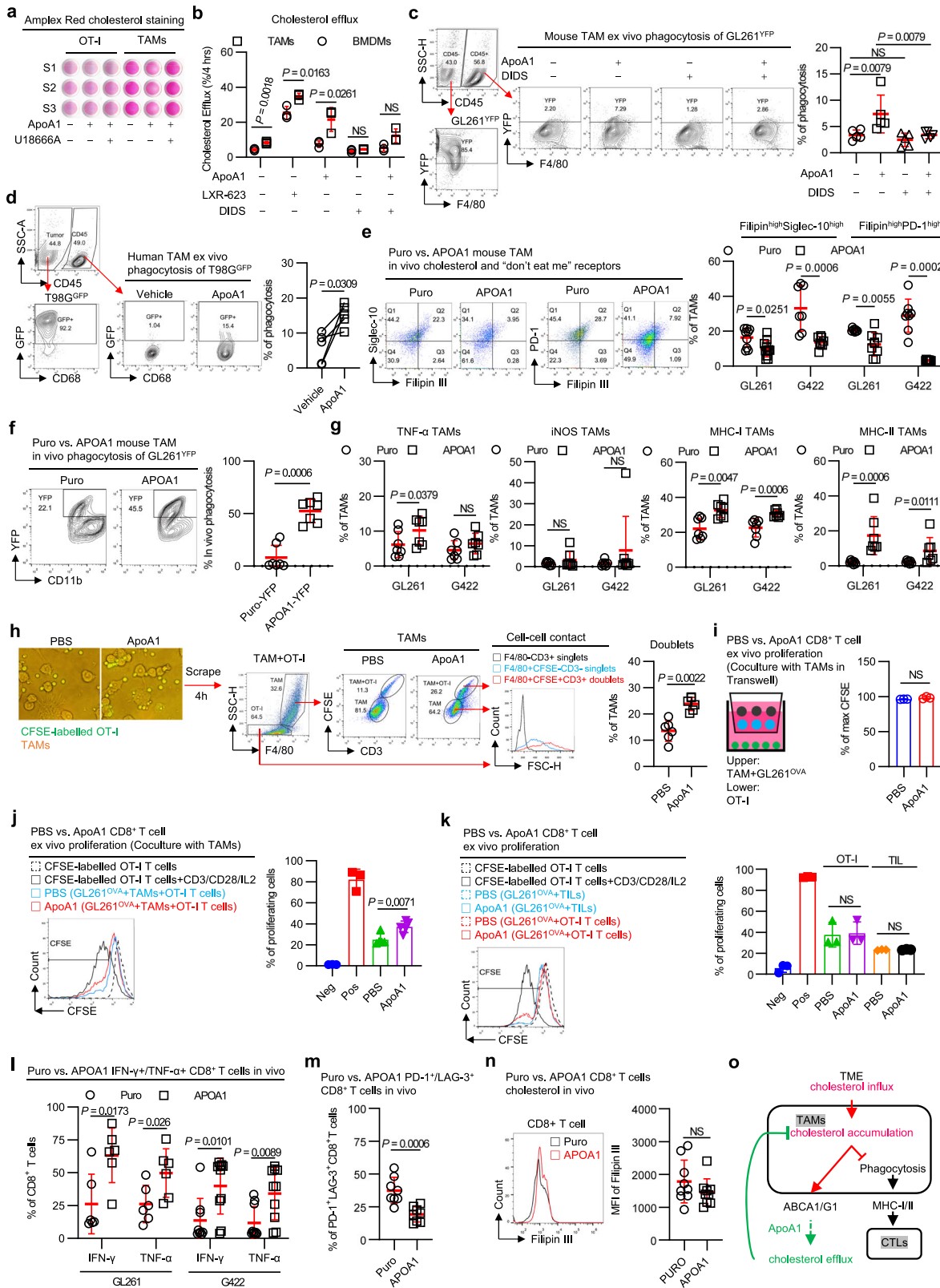

Our previous findings showed that ApoA1-mediated cholesterol efflux enhanced phagocytosis in TAMs. RNA-seq data indicated that intratumoral administration of AdV^APOA1 also upregulated phagocytosis-related genes relative to AdV^Ctrl treatment (Fig. 7o). In vitro viral infection GL261^RFP assays confirmed that AdV^APOA1 significantly facilitated TAM phagocytosis compared to AdV^Ctrl (Fig. 7p). In vivo studies demonstrated that AdV^APOA1 intratumoral administration reduced the expression of the phagocytosis-inhibitory receptor PD-1 on TAMs compared to AdV^Ctrl treatment (Fig. 7q) and increased antigen-presenting MHC-II in TAMs (Fig. 7r). Consistently, promoted lymphocyte infiltration and tumor clearance (Fig. 7s). Our findings suggest that an oncolytic adenovirus engineered to express APOA1 can reshape the GBM metabolic/immunological microenvironment.

**Fig. 4 | ApoA1-targeted cholesterol efflux reinvigorates TAM functional specialization. a** Amplex Red staining of cellular cholesterol levels. Data are related to the experimental data (Fig. 3h). Experiment conducted once. n = 3 biological samples. **b** Quantification of macrophage cholesterol efflux. $n = 3$ biological samples. (**c-d**) Quantification of ex vivo phagocytosis in mouse TAMs (**c**) and human TAMs (**d**). $n = 5$ biological samples. **e** Quantification of the expression of "don't eat me" receptors and cholesterol contents of TAMs in GBM mice expressing APOA1. GL261, $n = 11$ mice per group. G422, $n = 8$ mice per group. **f** Quantification of in vivo phagocytosis in TAMs in orthotopic GBM-bearing mice. $n = 7$ mice per group. **g** Quantification of antigen-presenting complex MHC-I/II and pro-inflammatory cytokine TNF-α/iNOS levels of TAMs in GBM mice expressing APOA1. $n = 7$ mice per group. **h** Determination of cell-to-cell contacts between TAMs and OT-I T cells. $n = 6$

biological samples. Ex vivo proliferation of CD8⁺ T cells cocultured with space-separated TAMs (**i**), space-contacted TAMs (**j**), and tumor cells (**k**). Separated TAMs, $n = 3$ independent experiments. Contacted TAMs, n = 6 independent experiments. Tumor cells, $n = 3$ independent experiments. (**l**) Quantification of cytotoxic INF-γ/TNF-α production in tumor-infiltrating CD8⁺ T cells in vivo. GL261, $n = 6$ mice per group. G422, $n = 10$ mice per group. Quantification of PD-1⁺LAG-3⁺ (**m**) and cholesterol levels (**n**) in tumor-infiltrating CD8⁺ T cells in GL261 tumors. PD-1⁺LAG-3⁺, $n = 8$ mice per group. Cholesterol levels, $n = 9$ mice per group. **o** Schematic diagram of ApoA1 acting on the TAM-T-cell axis. Statistical significance was determined using the Mann–Whitney test (Two-tailed) in **c, e, f, g, h, l, m, n**, the unpaired t test (Two-tailed) in **b, i, j, k**, or the paired *t* test (Two-tailed) in **d**. All data are the mean ± SD. Source data are provided in the Source Data file.

## Intratumoral administration of AdV^APOA1 initiates systemic antitumor immune responses

Next, we investigated the immune infiltration profile of orthotopic GBMs treated with AdV^APOA1. Our findings revealed that AdV^APOA1 treatment significantly increased T-cell infiltration relative to AdV^Ctrl treatment (Fig. 8a). Both AdV^APOA1 and AdV^Ctrl treatment increased PDCD1 ligand 1 (PD-L1) expression in tumors and TILs (Fig. 8b). AdV^APOA1 intratumoral administration boosted the effector T-cell phenotype, including CD44 expression and IFN-γ and granzyme B production (Fig. 8c), and reduced PD-1, LAG-3, and T-cell immunoreceptor with Ig and ITIM domains (TIGIT) expression on GL261 GBM-infiltrating CD8⁺ T cells (Fig. 8d). Similarly, in another orthotopic G422 GBM model, the intratumoral administration of AdV^APOA1 significantly reduced PD-1⁺LAG-3⁺ CD8⁺ T cells in the TME (Fig. 8e). These results suggest that AdV^APOA1 intratumoral administration enhances CD8⁺ T-cell-mediated tumor clearance.

Using a GO enrichment analysis network, we identified that the intratumoral administration of AdV^APOA1 initiated a systemic antitumor immune response, resulting in myeloid cell migration, T-cell costimulation, and IFN-γ production (Fig. 8f). To evaluate whether AdV^APOA1 could initiate systemic antitumor immunity, we established a bilateral subcutaneous GBM model (Fig. 8g). Our results showed that AdV^APOA1 intratumoral administration had more profound tumor control than AdV^Ctrl on both the virus-injected and virus-noninjected sides (Fig. 8h). Systemic tumor control was associated with CD8⁺ T-cell-mediated antitumor immunity (Fig. 8i), rather than viral shedding (Fig. 8j). Similarly, CD8⁺ T-cell or macrophage depletion abolished the antitumor activity of AdV^APOA1 (Fig. 8k), suggesting that AdV^APOA1 achieves tumor control by initiating a macrophage–T-cell axis.

## AdV^APOA1-cured mice establish long-term tumor-specific immune memory

The presence of M1-like TAMs promotes the tumor residency of memory T cells, which is associated with better immunotherapy outcomes[43]. We explored whether ApoA1-reprogrammed TAMs could facilitate tissue residency of T cells and establish tumor-specific immune memory. Our findings demonstrated that splenocytes isolated from the cured mice bearing subcutaneous GL261 GBM exhibited specific cytotoxicity towards GL261 cells but not MC38 cells (Supplementary Fig. 8a–c). Furthermore, cured mice with orthotopic GL261 GBM were resistant to subcutaneous rechallenge with GL261 but not MC38, suggesting that AdV^APOA1-induced immune surveillance was not limited to intracranial regions but was systemic (Supplementary Fig. 8d, e).

Similarly, intracranial rechallenging with GL261 did not result in tumors in the brains of cured mice (Supplementary Fig. 8f, g). Furthermore, we observed increased CD44⁻CD62L⁺ central memory CD8⁺ T cells and CD44⁺CD62L⁻ effector memory CD8⁺ T cells from cured mice (Supplementary Fig. 8h). Collectively, our findings demonstrate that the intratumoral administration of AdV^APOA1 can induce systemic tumor-specific immunity by reprogramming TAM cholesterol metabolism (Fig. 8l).

## Discussion

Immunotherapy holds promise for treating malignant GBMs. However, the anatomical, molecular, and metabolic specificities of GBMs are often intertwined to exert intrinsic and/or adaptive immunosuppressive mechanisms[44,45]. In this study, we characterized how cholesterol codependency shapes the immunosuppressive microenvironment in GBMs and proposed a therapeutic approach targeting TAM cholesterol efflux to enhance oncolytic viro-immunotherapy in GBMs. This study provides insights into the development of effective treatments for GBMs.

Characterizing the cholesterol metabolic status of GBMs is a prerequisite for precise targeting. Our study revealed that unlike extracranial tumors[46,47], GBM tissue had lower cholesterol levels than surrounding brain tissue and relied heavily on exogenous cholesterol uptake. This was evident from the higher expression of EGFR, which facilitates exogenous cholesterol uptake through LDLR, and the lower expression of SREBP2, a cholesterol sensor that regulates cholesterol synthesis via 3-hydroxy-3-methylglutaryl-coA reductase (HMGCR), in GBM tumor tissue compared to normal brain tissue[6,48]. Intriguingly, we also found that the cell-free cholesterol level was significantly higher in GBM TIFs, leading to the formation of cholesterol reservoirs in the TME and accumulation of cholesterol in tumor-infiltrating immune cells, particularly TAMs which exhibited the highest accumulation. These findings have important implications for developing targeted therapies for GBMs.

Targeting cholesterol metabolism in immune cells is a complex task that requires consideration of various spatiotemporal factors. The role of cholesterol in tumor-infiltrating immune cells remains controversial, with some studies suggesting that cholesterol accumulation promotes tumorigenesis while others propose that cholesterol deficiency leads to immune tolerance[10,12,46,47,49,50]. Recent research has shown that ovarian cancer cells secrete hyaluronidase to induce ABCA1/G1 expression in TAMs, leading to increased cholesterol efflux, decreased cholesterol levels, and an anti-inflammatory (M2) phenotype[12]. However, another study found that inhibiting cholesterol synthesis can promote TAM polarization from M2 to M1[51]. These discrepancies may be related to their distinct TMEs. Our study found that although GBM-infiltrating macrophages expressed high levels of ABCA1/G1, they still accumulated significant amounts of cholesterol, with levels 2–2.6-fold higher than those of peripheral macrophages. This suggests that GBMs have a cholesterol-rich TME, where high extracellular cholesterol potential in TIFs limits intracellular cholesterol efflux from TAMs, despite TAMs initiating decholesterolization feedback to upregulate ABCA1/G1 expression[52].

The impact of cholesterol accumulation on TAM functions in the GBM microenvironment is not well understood. Our study revealed that high cholesterol levels in the TME impaired the phagocytic activity of TAMs, possibly as a self-protective mechanism to limit cholesterol uptake. Given the limited ability of TAMs to relieve intracellular cholesterol stress via the ABCA1/G1 pathway in the high cholesterol microenvironment, reducing cholesterol at the source may be a more effective alternative. This is because phagocytic uptake and digestion

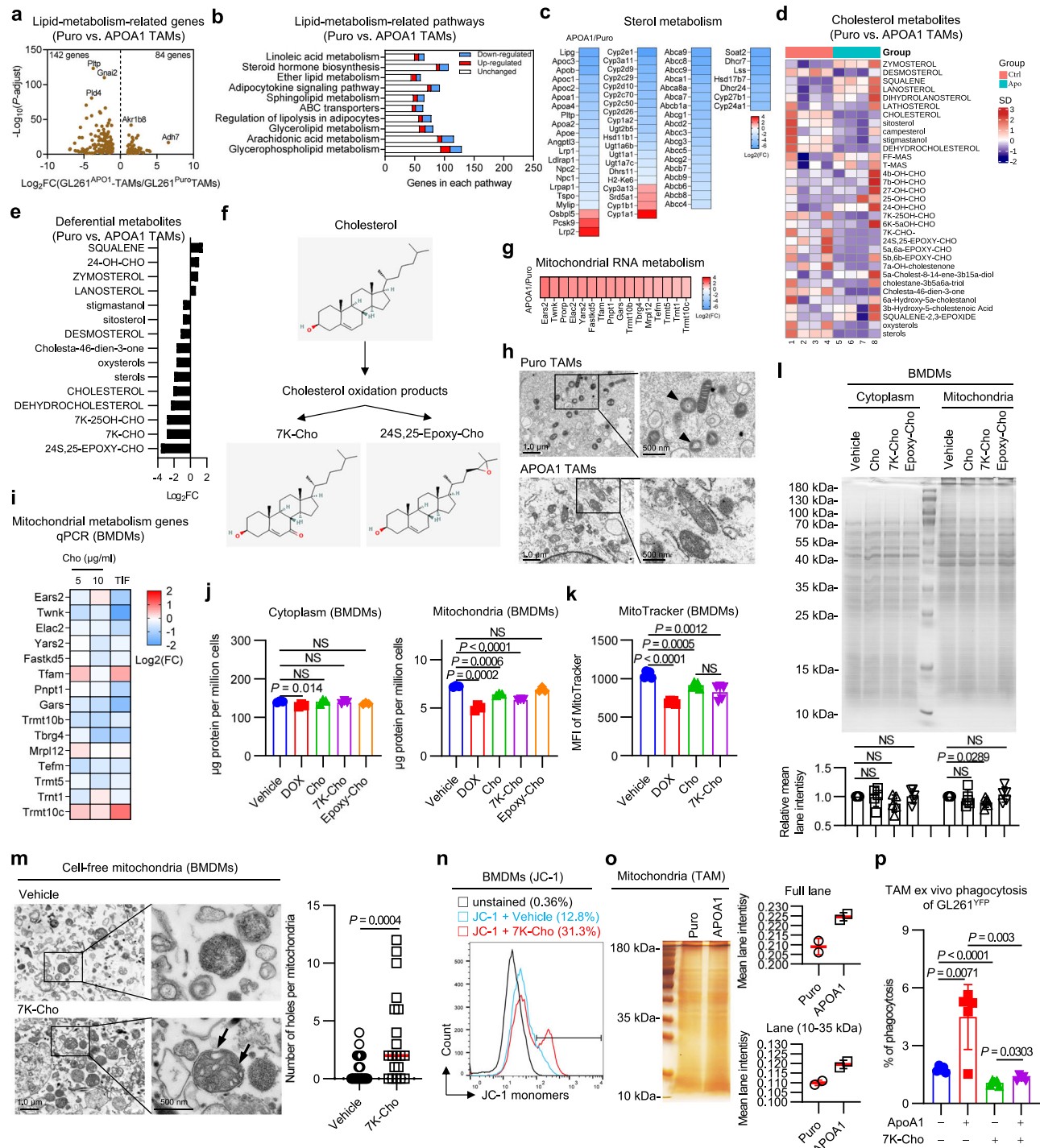

**Fig. 5 | ApoA1-enforced lipid remodeling reduces 7-ketocholesterol impairment of TAM mitochondrial translation.** Volcano plot (**a**), lipid metabolism-related pathway (**b**), and sterol metabolism heatmap (**c**) of differentially expressed genes of between Puro-TAMs and APOA1-TAMs. *n* = 3 biological samples per group. Heatmap (**d**) and quantification (**e**) of cholesterol-targeted metabolomics between Puro-TAMs and APOA1-TAMs. *n* = 4 biological samples per group. **f** Schematic molecular structure of cholesterol oxidation to 7-ketocholesterol (7K-Cho) and 24(S),25-epoxy-cholesterol (Epoxy-Cho). **g** Heatmap of differentially expressed mitochondrial RNA metabolism genes between Puro-TAMs and APOA1-TAMs. **h** Representative TME images of mitochondria of TAMs. Arrows indicate small mitochondria with less content. The images are representative of 2 samples per group. **i** qPCR analysis of mitochondrial RNA metabolism genes. *n* = 2 independent experiments. Quantification of cytoplasmic and mitochondrial proteins (**j**) and mitochondrial activity (**k**) in cholesterol-treated BMDMs. Protein, *n* = 3 biological samples per group. MitoTracker, *n* = 5 biological samples per group.

**l** Representative Coomassie staining of SDS–PAGE gel of cytoplasmic protein and mitochondrial protein of BMDMs. n = 5 independent experiments.
**m** Representative TME images of 7K-Cho-treated cell-free mitochondria and quantification of the number holes in mitochondria. Arrows show mitochondria with holes or loss of contents. The number of holes per mitochondria within a TEM image of a 1 μm size field of view was quantified. The images are representative of 3 samples. **n** Representative flow plots of the mitochondrial membrane potential of BMDMs. *n* = 2 biological samples per group. **o** Representative silver staining and quantification of SDS–PAGE gels of mitochondrial proteins in Puro-TAMs and APOA1-TAMs. n = 2 biological samples per group. **p** Quantification of ex vivo phagocytosis in ApoA1- and 7K-Cho-treated TAMs. n = 5 biological samples per group. Statistical significance was determined using the unpaired t test (Two-tailed) in **j**, **k**, **l**, **m**, **p**. All data are the mean ± SD. Source data are provided in the Source Data file.

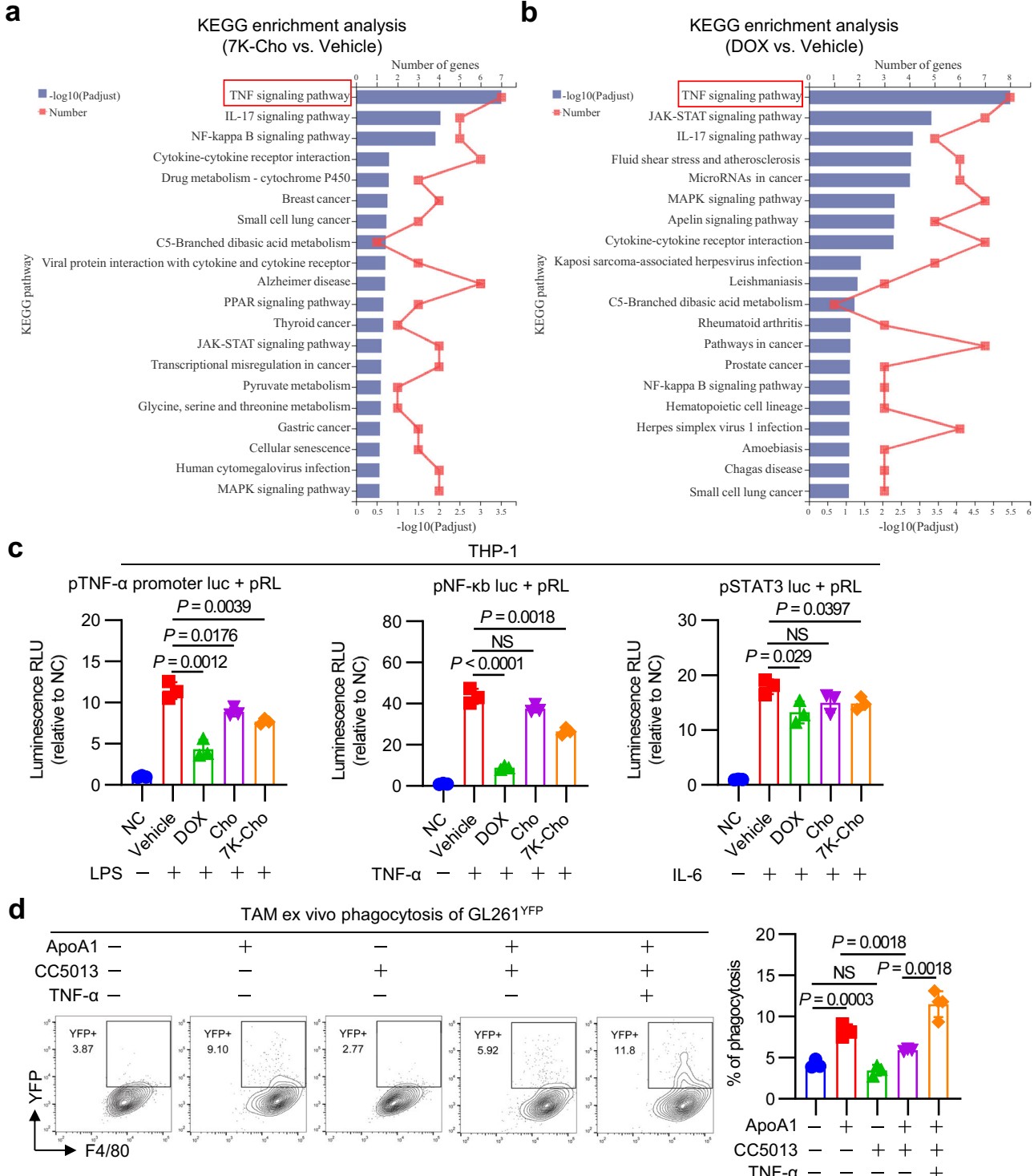

**Fig. 6 | Mitochondrial damage by 7-ketocholesterol induces dysregulation of the TNF signaling pathway in macrophages.** KEGG pathway enrichment analysis of differentially expressed genes between 7K-Cho-treated (**a**) or DOX-treated (**b**) and vehicle-treated BMDMs for 6 h. Enrichment analysis was performed using Fisher's exact test. In order to control the calculation of false positives, BH method was provided to correct the $P$ value (adjust P). $n = 3$ biological samples per group. **c** THP-1 cell luciferase reporter gene assays. THP-1 cells were co-transfected with the pTNF-α promoter luc plasmid, pNF-κb luc plasmid, pSTAT3 luc plasmid, or pRL-SV40 plasmid. After 24 h, the cells were harvested and cultured with 1% FBS medium containing vehicle, DOX (10 μg/ml), cholesterol (10 μg/ml), or 7K-Cho (10 μg/ ml) medium for another 24 h. Cells were stimulated with LPS (100 ng/ml), TNF-α (20 ng/ml), or IL-6 (10 ng/ml) for 6 h, and cells were lysed to detect firefly luciferase and Renilla luciferase. $n = 3$ independent experiments. **d** Quantification of ex vivo phagocytosis in ApoA1- and TNF-α-inhibitor (CC5013)-treated TAMs. TAMs isolated from intracranial GL261-bearing mice were cultured with vehicle, ApoA1 (10 μg/ml), or CC5013 (20 nM) medium for 24 h; or TNF-α (20 ng/ml) for 6 h. $n = 3$ biological samples per group. Statistical significance was determined using the unpaired t test (Two-tailed) in **c**, **d**. All data are the mean ± SD. Source data are provided in the Source Data file.

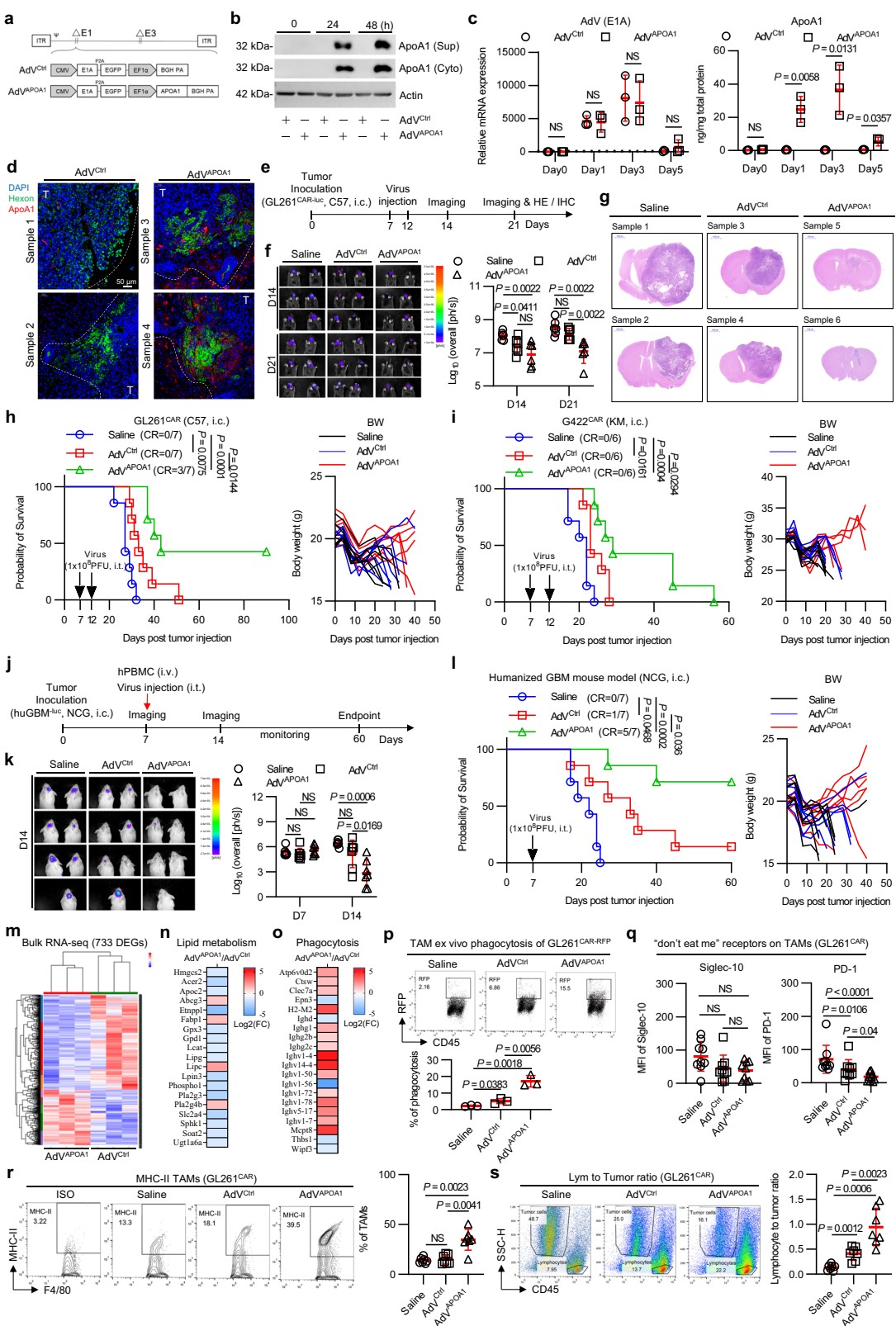

can increase intracellular cholesterol levels[53], which may explain why cholesterol accumulation inhibits phagocytosis in TAMs.

The inhibition of phagocytosis induced by high cholesterol levels in the TME may be linked to the upregulation of "don't eat me" receptors, such as Siglec-10 and PD-1. Additionally, SIRP-α is a phagocytosis checkpoint in macrophages that promotes tumor immune resistance[54]. Our study found that SIRP-α expression was lower in GBM

macrophages than in peripheral macrophages. Preclinical studies have shown that blocking CD47 with antibodies or oncolytic viruses can convert TAMs to a pro-inflammatory phenotype and enhance phagocytosis of tumor cells in GBM models[55,56]. However, the phagocytic fragility of TAMs in the TME may be complex, and the pattern and amount of "don't eat me" receptors co-expressed by the same TAMs may significantly affect the dysfunction severity. It has been suggested

**Fig. 7 | Oncolytic adenovirus expressing ApoA1 subverts TAM fragility and potentiates antitumor responses in GBMs. a** Construction of recombinant oncolytic adenoviruses (AdV[Ctrl] and AdV[APOA1]). **b** Western blot detection of ApoA1 protein expression in AdV[APOA1]-infected 293T cells. Culture supernatant and cell lysate were collected for identification of ApoA1 protein levels. *n* = 2 independent experiments. **c** In vivo replication and expression of AdV[APOA1]. *n* = 3 mice per group. (**d**) Representative immunofluorescence images of in vivo infection, replication and expression of AdV[APOA1]. The images are representative of 2 tumors per group. **e**–**g** Experimental setup (**e**), IVIS real-time monitoring (**f**), and representative H&E staining of tumor growth in vivo following intratumoral injection of oncolytic adenovirus in an immune-competent murine GL261[CAR-luc] model. *n* = 6 mice per group. Kaplan–Meier survival curves and body weights of intracranial GL261[CAR]-bearing C57BL/6 J mice (**h**) and intracranial G422[CAR]-bearing KM mice (**i**) that received treatment as indicated. GL261, *n* = 7 mice per group. G422, *n* = 6 mice per group. Experimental setup (**j**), tumor growth monitoring (**k**), and Kaplan–Meier survival curves and body weights (**l**) of an immune-reconstructive human GBM model treated with the indicated oncolytic adenoviruses. *n* = 7 mice per group. Heatmap of differentially expressed genes (**m**), lipid metabolism genes (**n**), and phagocytosis genes (**o**) in oncolytic virus-treated tumor tissues. A total of 733 DEGs were identified. *n* = 3 biological samples per group. **p** Quantification of ex vivo phagocytosis in TAMs. GL261[CAR-REP] cells were infected with saline, AdV[Ctrl] (MOI = 10) or AdV[APOA1] (MOI = 10) for 24 h. *n* = 3 biological samples per group. Quantification of the expression of the "don't eat me" receptor (**q**) and MHC-II (**r**) on TAMs. Siglec-10/PD-1, *n* = 9 mice per group. MHC-II, *n* = 7 mice per group. **s** The ratio of tumor-infiltrating lymphoid cells to tumor cells. *n* = 7 mice per group. Statistical significance was determined using the unpaired *t* test (Two-tailed) in **c**, **d**, **p**, the Mann–Whitney test (Two-tailed) in **f**, **k**, **q**, **r**, **s**, the log-rank test in **h**, **i**, **l**. All data are the mean ± SD. Source data are provided in the Source Data file.

that combining CD47 and CD24 antibodies may improve the overall efficacy of clinical treatment of GBMs[57]. However, given that acquired resistance and adverse events (e.g., acute anemia and thrombocytopenia) have been observed in systemic CD47/CD24 antibody administration[58,59], the feasibility of blocking CD47/CD24 alone or in combination in GBM remains uncertain. Our study proposes an alternative approach to downregulate "don't eat me" receptors by reducing cholesterol in TAMs, which may offer a therapeutic option for GBMs.

Mitochondrial dysfunction is another factor that affects phagocytosis by macrophages, and recent evidence suggests that it plays a crucial role in GBM progression[60]. However, the relationship between mitochondrial dysfunction and the TAM phenotype in the TME is not yet fully understood. Our study found that cholesterol accumulation impaired mitochondrial translation in TAMs, and the toxic oxysterol 7-ketocholesterol had a significant impact on mitochondrial function and phagocytosis in TAMs[61,62]. Therefore, cholesterol-induced mitochondrial impairment through the oxidative product 7-ketocholesterol could inhibit phagocytosis in TAMs. Our results demonstrated that ApoA1-mediated cholesterol efflux in TAMs effectively alleviated 7-ketocholesterol-induced mitochondrial impairment and restored phagocytosis in TAMs.

TAMs constitute up to 50% of GBM tissues and are known to promote gliomagenesis and resistance to immunotherapy[63]. However, studies have shown that depleting TAMs by CSF1R blockade failed to improve survival or resulted in high relapse rates[64,65]. In contrast, a preclinical study demonstrated that HSV armed with CD47 antibodies controlled GBM progression by enhancing phagocytosis in TAMs[56]. It is important to note that multiple factors contribute to the immunosuppressive properties of TAMs. Our study revealed that ABCA1/G1 expression was predominantly found in GBM tumor tissues, particularly in TAMs, making it an ideal target for manipulating cholesterol efflux. Indeed, enhancing cholesterol efflux and reducing cholesterol levels in TAMs restored their functional specialization, including phagocytosis and antigen presentation.

Targeting sterol metabolism has been a promising approach in GBM therapy. LXR-623 is a full agonist of LXR-β that can penetrate the blood–brain barrier and has been shown to have anti-GBM cell effects by promoting ABCA1/G1-mediated cholesterol efflux while reducing LDLR-mediated cholesterol uptake[6]. However, adverse CNS events observed in phase I clinical trials have hindered its practical application[8]. In the brain, ABC transporters are necessary for releasing cholesterol from astrocytes, while LDLR is essential for neurons to accept cholesterol-ApoE vehicles. Therefore, long-term dosing of LXR-623 in GBM patients may disrupt neuronal cholesterol homeostasis, leading to CNS toxicity. To avoid impairing neuronal cholesterol homeostasis, we employed an alternative strategy that targeted TAM cholesterol efflux via ApoA1 to control tumor immunity.

In this study, we developed a recombinant APOA1-expressing adenovirus for targeted manipulation of cholesterol metabolism in glioblastoma therapy. Intratumoral administration of AdV[APOA1] resulted in activation of the TAM-T-cell axis and downregulation of immune checkpoints, such as PD-1, LAG-3, and TIGIT. This approach also induced a systemic tumor-specific immune memory and demonstrated excellent safety in long-term toxicity assessments. Our findings suggest that oncolytic viruses may serve as a promising platform for developing metabolically-based cancer immunotherapies. Furthermore, another study showed that recombinant leptin expressing vaccinia virus enhanced antitumor immune responses by restoring mitochondrial metabolism in tumor-infiltrating CD8[+] T cells[66].

Our study has discovered a mechanism responsible for the phagocytic fragility of TAMs induced by GBM's metabolic microenvironment. It emphasizes the significance of regulating the TAM-T-cell immune codependency axis in GBM immunotherapy by manipulating cholesterol metabolism. Moreover, we propose an immunometabolic therapeutic approach targeting cholesterol efflux in TAMs through OV-targeted ApoA1 delivery, which has been proven effective and safe for GBM treatment and can be translated to other preclinical or clinical cancer therapies.

Our research indicates that GBM metabolism causes cholesterol accumulation in TME, leading to TAM phagocytic fragility. However, there are still some limitations to this study. Although we observed that 7K-Cho disrupts TAM mitochondrial translation, the precise mechanisms behind this effect require further investigation.

## Methods
### Experimental model and subject details
Four-to-six-week-old male Balb/c-Nude, six-to-eight-week-old male C57BL/6J mice, NOD/ShiLtJGpt-Prkdc[em26Cd52]Il2rg[em26Cd22]/Gpt (NCG) mice, and B6/JGpt-Ptprc[em1Cin(p. K302E)]/Gpt (CD45.1) mice were purchased from the Model Animal Research Center of Nanjing University. Six-to-eight week old male C57BL/6-Tg (TcraTcrb) 1100Mjb/J (OT-I) mice were purchased from the Jackson Laboratory. Six-to-eight week old male KM mice and SD rats were purchased from Nanjing Junke Biotechnology Co., Ltd, China. Animals were housed in SPF facilities at the Medical School of Nanjing University. Housing conditions were as follows: the light time was 8:00–20:00, the temperature was 18–22 °C, and the relative humidity was 40–70%. Animal care and handling procedures were in accordance with the NIH Guide for the Care and Use of Laboratory Animals and were approved by the Nanjing University Institutional Review Board.

Intracranial models were established as previously described;[67,68] briefly, $1 \times 10^5$ GBM cells or were stereotactically inoculated into the caudate nucleus of the brain (for mouse, 3.0 mm depth to the skull 1.0 mm anterior and 2.0 mm lateral to bregma; for rat, 5.0 mm depth to the skull 1.0 mm anterior and 3.0 mm lateral to bregma). For example, C57BL/6 J mice were inoculated i.c. with $1 \times 10^5$ GL261 cells; KM mice were inoculated i.c. with $1 \times 10^5$ G422 cells; Balb/c-Nude mice were inoculated i.c. with $1 \times 10^5$ U251-MG cells; SD rats were inoculated i.c. with $1 \times 10^5$ C6 cells. Animals were monitored for survival and neurological symptoms every other day. The mice were sacrificed either when

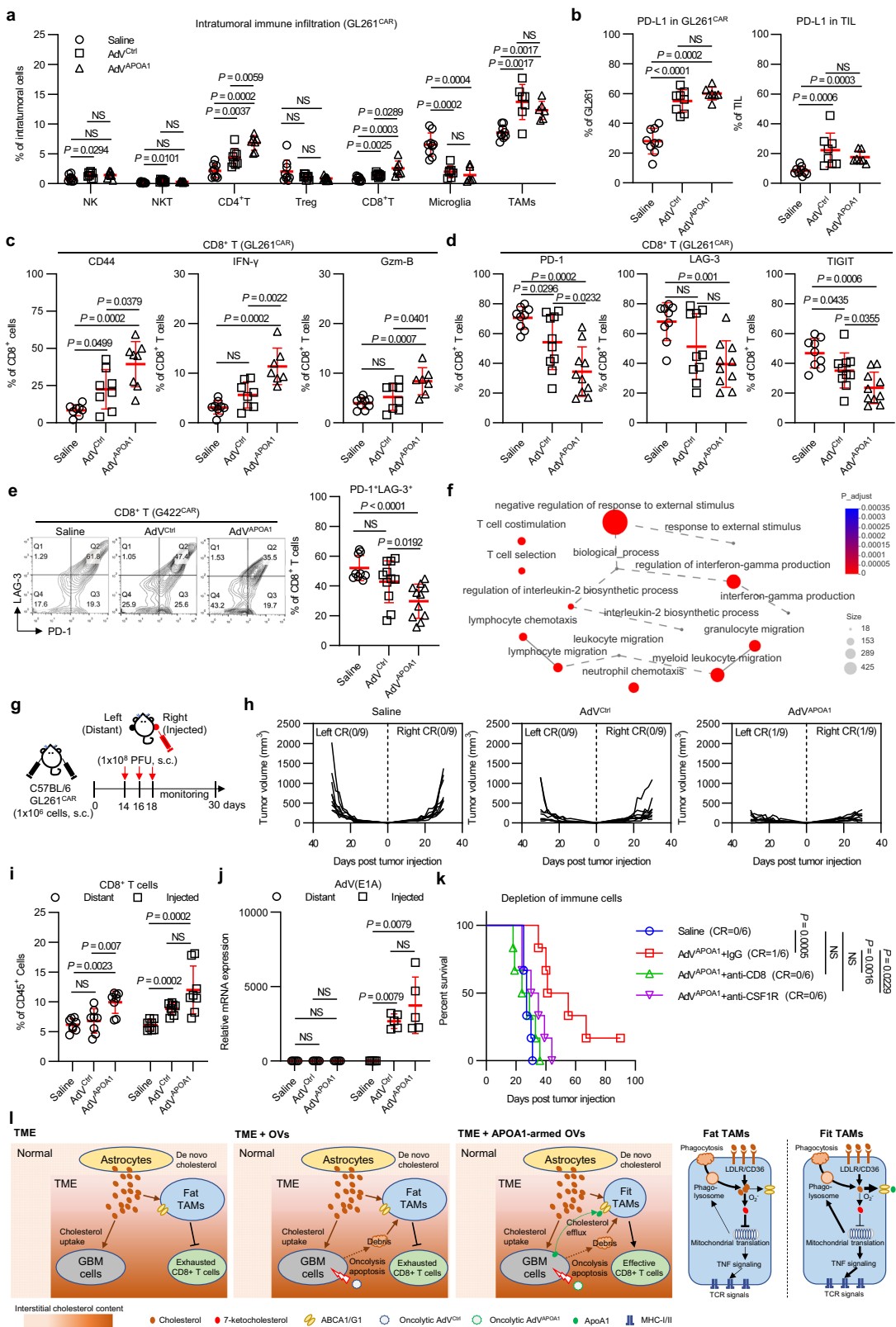

they became moribund or exhibited a significant weight loss (>20%). Tumor or normal tissue samples were collected for further analysis.

Adoptive transfer of BMDMs was performed as previously described;[69] briefly, in vitro differentiated CD45.1+ BMDMs were cultured with vehicle or cholesterol-containing medium (10 µg/ml) for 2 days. Cells were harvested and transferred into 14-day GL261YFP-bearing wild-type CD45.2+ C57BL/6J mice intravenously once daily for

three days. One day after the last transfer, the mice were sacrificed and CD45.1+ and CD45.2+ TAMs were examined for in vivo phagocytosis by flow cytometry. For analysis of "don't eat me receptors", CD45.1+ splenic macrophages and TAMs were examined for cholesterol content and Siglec-10 and PD-1 expression by flow cytometry.

For immune cell depletion, C57BL/6J mice were inoculated i.c. with $1 \times 10^5$ GL261Puro cells or $1 \times 10^5$ GL261APOA1 cells, respectively.

**Fig. 8 | Intratumoral administration of oncolytic AdV$^{APOA1}$ elicits systemic antitumor immune responses.** Quantification of intratumoral infiltrating immune cell subsets (**a**), PD-L1 expression in tumor cells (**b**), CD44/IFN-γ/Gzm-B expression in CD8$^+$ T cell (**c**), and PD-1/LAG-3/TIGIT immune checkpoint expression in CD8$^+$ T cells (**d**). Immune cells, $n = 9$ mice per group. PD-L1, $n = 9$ mice per group. CD44/IFN-γ/Gzm-B, $n = 10$ mice per group. PD-1/LAG-3/TIGIT, $n = 10$ mice per group. **e** Quantification of PD-1$^+$LAG-3$^+$ exhausted tumor-infiltrating CD8$^+$ T cells. Saline, $n = 9$ mice. Virus, $n = 11$ mice. **f** Network diagram of GO enrichment analysis of differentially expressed genes in AdV$^{Ctrl}$- and AdV$^{APOA1}$-treated tumor bulks. Data are related to the experimental data (Fig. 7m). Experimental setup (**g**) and tumor growth curves (**h**) of the subcutaneous contralateral GBM model. $n = 9$ mice per group. Quantification of tumor-infiltrating CD8$^+$ T cells (**i**) and viral replication (**j**) in the subcutaneous bilateral model. CD8$^+$ T cell, $n = 8$ mice per group. Virus replication, $n = 5$ samples per group. **k** Kaplan–Meier survival curves of AdV$^{APOA1}$-treated GL261$^{CAR}$-bearing mice with or without depletion of immune cells. Intracranial GL261$^{CAR}$-bearing C57BL/6J mice were injected i.p. with isotype IgG, anti-CD8, or anti-CSF1R antibodies on days 2, 7, 12 and 17 after tumor inoculation. Mice received i.t. injection of saline or $1 \times 10^8$ PFU AdV$^{APOA1}$ on day 7 and day 12 after tumor inoculation. Survival was monitored every day. $n = 6$ mice per group. **l** Schematic diagram of macrophage plasticity in a cholesterol-rich microenvironment. Cholesterol accumulation leads to phagocytic fragility in TAMs by inhibiting mitochondrial translation via 7-ketocholesterol. By manipulating cholesterol efflux through ApoA1, we were able to improve TAM function and enhance TAM-T-cell-mediated antitumor responses. Additionally, intratumoral administration of ApoA1-armed oncolytic adenovirus was found to remodel the immunometabolic microenvironment and promote antitumor immune responses. Statistical significance was determined using the Mann–Whitney test (two-tailed) in **a**–**e**, **i**, **j**, or the log-rank test in **k**. All data are the mean ± SD. Source data are provided in the Source Data file.

GL261$^{APOA1}$-bearing C57BL/6J mice were then treated with immune depletion agents, including 10 mg/kg isotype IgG, 10 mg/kg anti-NK1.1, 10 mg/kg anti-CD4, 10 mg/kg anti-CD8, and 25 mg/kg or 50 mg/kg clodronate, on the 5th day before the tumor transplantation, every 5 days for 20 days.

Rhesus monkeys were purchased from Hubei Tianqin Biotechnology Co., Ltd. Suizhou Branch (1)/Xinye County Xinyu Wildlife Breeding Co., Ltd (2). Animal certificate number: 421218300000162(1)/410934211100000964(2)/410934211100001081(2). Monkeys were housed in InnoStar GLP laboratory according to NMPA (CFDA) guidelines. The study protocol was approved by the Institutional Animal Care and Use Committee of Shanghai InnoStar Bio-Tech Co. Ltd (IACUC No.: IACUC-2021-M-072; 21091RD02). Both female and male animals were used in the preclinical safety studies.

The study involved patients with newly diagnosed primary brain cancer who underwent intracranial surgery for tumor resection and provided consent to donate the resected tissue. A part of the fresh tissue was used for pathology and molecular diagnosis, and the remaining tissue was used for the preparation of single-cell suspension. Finally, a total of 6 GBM patient tumor tissues were collected for analysis. The patients' clinical information can be found in Supplementary Table 1. This study has received approval from the Human Research Ethics Committee of the Second Affiliated Hospital of Zhejiang University School of Medicine (ID:2023-0261) and written informed consent was obtained from all participants included in the study. This study performed in accordance with the 1975 Declaration of Helsinki. An abstract (translated version) of the approved study protocol has been submitted as Supplementary Note 1.

## Cell culture and agents
Bone marrow-derived macrophages (BMDMs) were obtained as previously described[25]. Briefly, bone marrow was flushed from the femurs and tibia of 8- to 10-week-old C57BL/6J mice. Bone marrow cells were passed over 70 μm cell strainers and further resuspended in ACK lysis buffer at room temperature for 5 min to remove blood cells. The remaining cells were cultured at a density of $10^6$ cells/ml in complete DMEM containing M-CSF (10 ng/ml) for 7 days. OT-I CD8$^+$ T cells were obtained as previously described[10]. Briefly, CD8$^+$ T cells were isolated from the spleen of OT-I mice and cultured at a density of $10^6$ cells/ml in complete RPMI 1640 containing CD3ε/CD28 antibodies and IL-2 (10 ng/ml) for 5 days.

No cell lines listed in the ICLAC database were used. Commercial cell lines, including 293T (RRID: CVCL_0063, ATCC CRL-3216), U138-MG (RRID: CVCL_0020, ATCC HTB-16), T98G (RRID: CVCL_0556, Procell CL-0583), U251-MG (RRID: CVCL_0021, CCTCC GDC0093), THP-1 (RRID: CVCL_0006, CCTCC GDC0100), GL261 (RRID: CVCL_Y003, BCRJ 0299), G422 (NICR PUMC000314), MC38 (RRID: CVCL_B288, NICR PUMC000523), C6 (RRID: CVCL_0194, NICR PUMC000131) were cultured in DMEM medium (Gibco) supplemented with 10% FBS and 100 U/ml penicillin and streptomycin. Human cell lines were authenticated by short tandem repeat (STR) analysis. Mycoplasma contamination was tested by Myco-Lumi™ (Beyotime) monthly. Stable cell lines such as GL261$^{Puro}$, GL261$^{APOA1}$, G422$^{Puro}$, G422$^{APOA1}$, GL261$^{OVA}$, GL261$^{CAR}$, G422$^{CAR}$, GL261$^{luc}$, U138-MG$^{luc}$, T98G$^{GFP}$, and GL261$^{RFP}$ were generated by infecting 50% confluent cells with lentivirus and selecting with 1 μg/ml or 10 μg/ml puromycin according to the manufacturer's instructions. Supplementary Tables 2 and 3 contain a list of key resources and a list of antibody information.

## BIF and TIF collection
Three intracranial GBM models were established, including U251-MG human GBM in Balb/c nude mice, GL261 murine GBM in C57BL/6J mice, and C6 rat GBM in SD rats. Mice were sacrificed on days 19 to 23 after tumor implantation; rats were sacrificed on day 15. Normal brain interstitial fluid (BIF) and tumor interstitial fluid (TIF) were collected as previously described[13,46]. Briefly, fresh brain and end-stage tumor tissues were collected from tumor-bearing mice. PBS washed tissue. Equal weight tissues were dissected and placed on 20 μm cell strainers fixed on top of 50 ml conical tubes and centrifuged at $100 \times g$ for 10 min at 4 °C. The interstitial fluid adhered to the tube wall was washed with 100 μl PBS. The bottom collected liquid was further centrifuged at $10,000 \times g$ for 10 min at 4 °C to remove any tissue debris. The supernatants (BIF or TIF) were stored at −80 °C until further analysis.

## Cholesterol measurement
For cholesterol quantification analysis, cholesterol was extracted from fresh tissue or interstitial fluid using the Cholesterol Extraction kit (Sigma) and measured using the Amplex Red Cholesterol Assay kit (Invitrogen). For cell-level cholesterol analysis, in vitro activated OT-I T cells and TAMs sorted from GL261$^{OVA}$ tumors were seeded in 6-well plates and cultured with 1% FBS medium containing vehicle, ApoA1 (10 μg/ml), and/or U18666A (cholesterol transport inhibitor, 5 μM) for 48 h. Cells were then harvested for cholesterol content analysis. For tissue cholesterol analysis, total cholesterol levels were normalized to tissue weight. HDL and LDL/VLDL cholesterol levels in TIFs were measured using the HDL and LDL/VLDL Quantification Colorimetric/Fluorometric Kit (BioVision).

Cellular cholesterol efflux was measured using the Cholesterol Efflux Fluorometric Assay Kit (BioVision). TAMs and BMDMs were seeded in 96-well plates and incubated with serum-free cholesterol labeling medium for 1 h. Cholesterol efflux levels were tested after 4 h of incubation of these macrophages with serum-free vehicle, ApoA1 (10 μg/ml), or DIDS (ABCA1 inhibitor, 5 μM) medium.

For cholesterol staining, cells were stained with Filipin III or NBD cholesterol for flow cytometry detection according manufacturer's instructions.

## Flow cytometry

Brain tumor samples were minced in 2% RPMI 1640 medium containing 0.5 mg/ml collagenase type IV and digested at 37 °C for 30 min. Cells were further separated with 70 μm cell strainers and resuspended in excess of complete RPMI 1640 medium. Cells were centrifuged at $400 \times g$ for 5 min to obtain a single-cell suspension. Before staining, the cell suspensions were incubated with a 1: 100 diluted Fc blocker (anti-CD16/32) at 4 °C for 20 min. The cell suspensions were then stained with PI (5 μg/ml) at room temperature for 5 min to determine cell viability.

For cell surface staining, cell suspensions were incubated with the indicated antibodies (0.25 μg/$10^6$ cells) in 100 μl FACS buffer at 4 °C for 30 min. Tumor cells ($CD45^-$), monocytes ($CD11b^+F4/80^-$), macrophages ($CD11b^+F4/80^+$), microglia ($CD45^-CD11b^+$), and lymphoid cells ($CD45^+CD3^+$) were stained with Filipin III followed by flow cytometry analysis of cholesterol content. BMDMs were treated with vehicle and 7K-Cho (10 μg/ml) in complete medium for 6 h. The mitochondrial membrane potential of these cells was determined by JC-1 monomers.

For intracellular staining, PMA/ionomycin (eBioscience, 00-4970-93) and brefeldin A (MCE, HY-16592) was added to the cultures for 3-5 h to stimulate cytokine production and block cytokine secretion. Cells were harvested and followed by cell surface staining. Stain cells were centrifuged at $350 \times g$ for 5 min for removing the supernatant. Subsequently, 100 μl cell suspension was fixed with 150 μl Cyto-Fast™ Fix/Perm buffer at 4 °C for 30 min. Cells were washed twice with Cyto-Fast™ Perm Wash solution and further incubated with intracellular antibodies in 100 μl Perm buffer at 4 °C for 30 min. Wash the cells with 1 ml Cyto-Fast™ Perm Wash solution. Resuspend the cells in 300 μl of Cell Staining Buffer/PBS.

For CFSE staining, T cells were incubated with 5 μM CFSE working solution for 20 min at room temperature. In cell-cell contact assay, TAMs were pretreated with vehicle or ApoA1 (10 μg/ml) in 1% FBS medium for 48 h. Then, TAMs were cocultured with CFSE-labeled OT-I T cells for another 4 h. Cells were scraped for cell contact analysis by flow cytometry. In T-cell division assay, CFSE-labeled T cells were added to the lower chambers; TAMs were cocultured with $GL261^{OVA}$ cells (1:1) in the upper chambers of a 3-μm-pore Transwell. Cells were further cultured with either vehicle or ApoA1 (10 μg/ml) in 1% FBS medium in 6-well plates for another 48 h. Cells were incubated with 10% FBS medium for 10 min to quench staining. T-cell proliferation was determined by flow cytometry.

Samples were analyzed using BD FACS Calibur, BD Aria I, Beckman CytoFLEX LX, ACEA Novacyte, or Gaugene MoniSight 820 and data were analyzed with FlowJo v10.8.1.

## Phagocytosis

Phagocytosis analysis of macrophages was performed as previously described[21]. Briefly, for ex vivo phagocytosis, in vitro differentiated BMDMs, anti-F4/80 sorted mouse TAMs, or anti-CD14 sorted human TAMs were plated in a U-bottom ultralow attachment 96-well plate at 37 °C for 20 min to allow them to rest after the indicated treatments in 1% FBS medium for 24-48 h.

For in vitro phagocytosis of NBD cholesterol-labeled tumor debris assay, GL261 cells were cultured with NBD cholesterol (10 μg/ml) for 2 days. A total of $1 \times 10^6$ cells was harvested and freeze−thawed to obtain cell debris. TAMs or BMDMs were cocultured with this debris for 4 h. Phagocytosis of cell debris was determined by flow cytometry.

For in vitro phagocytosis of S. aureus, BMDMs were incubated with medium containing the indicated concentrations of TIF (20 mg tumor in 1 ml medium, filtered), or β-CD (0.5 mM)-coated TIF in 1% FBS medium for 2 days. These macrophages were cocultured with $HI^+$ S. aureus (MOI = 10) for 30 min. Phagocytosis was calculated by measuring the number of $F4/80^+HI^+$ by flow cytometry.

For BMDM ex vivo phagocytosis of tumor cells, BMDMs were incubated with medium containing vehicle, cholesterol (10 μg/ml), and LXR-623 (5 μM) in 1% FBS medium for 48 h. These BMDMs were harvested and cocultured with $GL261^{YFP}$/$T98G^{GFP}$ (1:2) cells in a U-bottom ultralow attachment 96-well plate in complete medium for another 4-12 h at 37 °C. All cells were then harvested and incubated with cold Trypsin-EDTA at room temperature for 10 min to detach surface adhesion.

For TAM ex vivo phagocytosis of tumor cells, TAMs were isolated by anti-F4/80 or anti-CD14 MicroBeads and cultured with 1% FBS medium containing vehicle, ApoA1 (10 μg/ml), DIDS (5 μM), or 7K-Cho (10 μg/ml) for 48 h. Subsequently, TAMs were cocultured with $GL261^{YFP}$ or $T98G^{GFP}$ cells in complete medium for another 4 h using a U-bottom ultralow attachment 96-well plate. Phagocytosis was calculated by measuring the number of $F4/80^+YFP^+$ or $F4/80^+GFP^+$ macrophages.

For in vivo phagocytosis, intracranial $GL261^{Puro-YFP}$- or $GL261^{APOA1-YFP}$-bearing C57BL/6J mice were sacrificed on day 17 after tumor inoculation. In vivo phagocytosis of TAMs was examined by measuring the number of $CD45^+CD11b^+F4/80^+YFP^+$ macrophages.

## Luciferase Reporter Assay

For in vitro assay, naïve and OT-I T cells were cultured with OVA 254-264 (1 μg/ml) and IL-2 (10 ng/ml) for 5 days to acquire activated T cells. OVA or CAR-3xFlag sequences were subcloned into pLenti-SV40-Luciferase-IRES-Puro (Genechem). TAMs were isolated from orthotopic $GL261^{OVA}$, and then cells were mixed with $GL261^{OVA-luc}$ in the presence or absence of either naïve T cells or OT-I T cells at a ratio of 1:1:1. The mixture was further cultivated with medium supplemented with vehicle or ApoA1 (10 μg/ml) for another 48 h. $GL261^{OVA-luc}$ alone was used as a negative control. Cytotoxicity was examined by measuring luciferase activity with D-Luciferin according to the manufacturer's instructions (BioVision).

For in vivo assay, intracranial $GL261^{CAR-luc}$-bearing C57BL/6J mice received intratumor (i.t.) injection of equal volumes of saline, $1 \times 10^8$ PFU $AdV^{Ctrl}$, or $1 \times 10^8$ PFU $AdV^{APOA1}$ on days 7 and 12 after tumor inoculation. Tumor growth was quantified by IVIS on days 14 and 21 after tumor inoculation. Intracranial $U138-MG^{luc}$-bearing NCG mice were i.v. injected $2 \times 10^6$ human PBMCs and were further i.t. injected with saline, $1 \times 10^8$ PFU $AdV^{Ctrl}$, or $1 \times 10^8$ PFU $AdV^{APOA1}$ on day 7 after tumor inoculation. Tumor growth was identified by IVIS (LB 983 NC100) on days 7 and 14 after tumor inoculation. Data were analyzed by IndiGo (Version 2.0.5.0).

## RNA sequencing

TAMs (GSE201975) of $GL261^{Puro}$ or $GL261^{APOA1}$ tumors were collected by anti-F4/80 MicroBeads. In vitro vehicle-, DOX-, or 7K-Cho-treated BMDMs (GSE214907) were harvested. For tumor bulks (GSE201976) sequencing, intracranial 14-day $GL261^{CAR}$-bearing C57BL/6J mice were i.t. injected with $1 \times 10^8$ PFU $AdV^{Ctrl}$ or $1 \times 10^8$ PFU $AdV^{APOA1}$. The tumor bulks were then collected to perform transcriptome RNA-seq analysis 3 days after virus injection. Total RNA of these samples was extracted using TRIzol (Vazyme) and stored at −80 °C. Transcriptome analysis of these samples was performed using the Illumina NovaSeq 6000 sequencing platform (Majorbio).

## Cholesterol-targeted metabolomics

Cholesterol-targeted metabolomics was performed to determine changes in cholesterol and cholesterol metabolites of TAMs. Lipids in sorted TAMs were extracted by a modified Bligh/Dyer method. Cholesterol and its metabolites were detected by Exion UPLC QTRAP 6500 PLUS Sciex) LC/MS in electrospray ionization (ESI) mode (LipidALL Technologies). In brief, 500 μl of ethanol containing 5 μg of BHT was added to the cell. Internal standard cocktail (50 μl) comprising d6-lanosterol, d6-zymosterol, d7-desmosterol, d7-lathosterol, d7-d-dehydrocholesterol, d6-cholesterol, d7-24-hydroxcholesterol, d7-7β-hydroxycholesterol, d6-25-hydroxycholesterol, d7-7-ketocholesterol, d7-7α-hydroxy-cholestenone, and d6-TMAS. (Avanti Polar Lipids) was

added to the samples. The samples were incubated at $100 \times g$ for 15 min at 4 °C. At the end of the incubation, 250 µl of Milli-Q water and 1 ml of n-hexane were added.

The samples were mixed thoroughly by vortexing and then centrifuged at $13,800 \times g$ for 5 min at 4 °C. The clear upper phase containing oxysterols and sterols in hexane was transferred to a new tube. The extraction was repeated once with another 1 ml of n-hexane. The pooled extract was dried in a SpeedVac under organic mode. Oxysterols and sterols were derivatised to obtain their picolinic acid esters prior to LC/MS analysis and quantitated by referencing the spiked internal standards as previously described[70]. Heatmap showing the relative abundance of each cholesterol species normalized to protein concentration (shown as Z-row scores).

### Transmission electron microscopy

Anti-F4/80 MicroBeads-isolated mouse TAMs, anti-CD14 MicroBeads-isolated human TAMs, and vehicle or 7K-Cho-treated BMDMs were fixed in 2.5% glutaraldehyde at 4 °C for 30 min. Mitochondria were isolated from BMDMs and cell-free treated with vehicle and 7K-Cho (10 µg/ml) in complete medium for 6 h. The samples were shipped on ice and processed by Servicebio based on previously reported techniques[21]. Images were acquired utilizing the HITACHI HT 7800 120 kv and JEM-2100 Plus.

### Mitochondria quality and silver staining

For analysis of mitochondrial RNA metabolism, BMDMs were obtained and treated with cholesterol at a concentration of 5 or 10 µg/ml, or with TIF for 6 h. Then, samples were harvested and subjected to qPCR analysis of mitochondrial RNA metabolism genes (Supplementary Table 2).

For quantification of cytoplasmic proteins and mitochondrial proteins, BMDMs were cultured with vehicle, DOX (doxycycline, a mitochondrial translation inhibitor, 10 µg/ml), cholesterol (10 µg/ml), 7K-Cho (10 µg/ml), or Epoxy-Cho (10 µg/ml) in complete medium for 6 h. Then, the cells were harvested and counted, followed by mitochondrial separation from equal counts of BMDMs. Isolated mitochondria were further quantified by the Mouse mtDNA Probe PCR Kit (BJBALB, BTN15-141). Protein levels in the cytoplasm and mitochondria were quantified by BCA. Lysed cytoplasm and mitochondria proteins were subjected to SDS–PAGE for Coomassie Brilliant Blue staining.

For analysis of mitochondrial activity, BMDMs were treated with vehicle, DOX (10 µg/ml), cholesterol (10 µg/ml), or 7K-Cho (10 µg/ml) in complete medium for 6 h. The mitochondrial activity of these cells was determined by MitoTracker levels.

For TAMs were isolated from GL261[Puro] or GL261[APOA1] tumors. The same number of TAMs was subjected to mitochondrial isolation using the Cell Mitochondrial Isolation Kit (Beyotime). Mitochondria were further quantified by the Mouse mtDNA Probe PCR Kit (BJBALB, BTN15-141). Equal amounts of mitochondria were lysed and subjected to SDS–PAGE for silver staining (Beyotime). The average optical density of the lanes was calculated by ImageJ.

### Oncolytic adenoviruses

Oncolytic AdV[Ctrl] and AdV[APOA1] were generated as previously described[71]. Briefly, target sequences containing viral *E1A* (GenBank: MH629744.1) and Homo sapiens *APOA1* (GenBank: NC_000011.10) were fully synthesized by GeneScript and subcloned into the pShuttle (pENTER/D-TOPO) plasmid (Fig. 7a). The ENTR plasmid was further recombinant with the pAd/PL-DEST (Invitrogen) human adenovirus type 5 backbone to generate a recombinant adenovirus expression vector. These expression vectors were linearized by Pac I digestion and further transfected into 293T cells for virus generation. The oncolytic viruses were identified, amplified, purified, and titered according to our previous method[71].

For in vivo replicating and expression, intracranial GL261[CAR]-bearing C57BL/6J mice were i.t. injected with $1 \times 10^8$ PFU AdV[Ctrl] or $1 \times$

$10^8$ PFU AdV[APOA1] on day 14 after tumor inoculation. Tumor tissues were collected on days 0, 1, 3, and 5 post injection. Virus copy was determined by *E1A* gene levels by qPCR (Supplementary Table 2). ApoA1 protein expression was determined by ELISA.

For in vivo immune cell infiltration, intracranial GL261[CAR]-bearing C57BL/6J mice or G422[CAR]-bearing KM mice received i.t. injection of saline, $1 \times 10^8$ PFU AdV[Ctrl], or $1 \times 10^8$ PFU AdV[APOA1] on day 14 after tumor inoculation. Three days later, tumor tissues were collected, and single-cell suspensions were subjected to flow cytometry to determine the ratio of tumor-infiltrating lymphoid cells to tumor cells. The cell subpopulations were determined by flow cytometry using NK1.1$^+$CD3$^-$ for NK cells, NK1.1$^+$CD3$^+$ for NKT cells, CD4$^+$ T cells, Foxp3$^+$CD4$^+$ for regulatory T ($T_{reg}$) cells, CD8$^+$ T cells, CD45$^-$CD11b$^+$ for microglia, and CD11b$^+$F4/80$^+$ for TAMs.

In contralateral subcutaneous models, a total of $1 \times 10^6$ GL261[CAR] cells were inoculated s.c. into the right and left flanks of C57BL/6J mice. Then, the mice received i.t. injection of saline, $1 \times 10^8$ PFU AdV[Ctrl], or $1 \times 10^8$ PFU AdV[APOA1] on the right flanks on days 14, 16, and 18 after tumor inoculation. An equal volume of saline was injected i.t. into the left flanks of the same individual. Tumor growth was determined every other day. Tumor volume was calculated as $0.5 \times \text{length} \times \text{width}^2$. Mice were sacrifice when maximal tumor burden reached 2,000 mm³.

For bilateral tumor-infiltrating CD8$^+$ T cells and viral replication, the mice received i.t. injection of saline, $1 \times 10^8$ PFU AdV[Ctrl], or $1 \times 10^8$ PFU AdV[APOA1] on the right flanks on day 17 after tumor engraftment. An equal volume of saline was injected i.t. on the left flanks 3 days later. Then, tumor tissues were collected, tumor-infiltrating CD8$^+$ T cells were determined by flow cytometry and viral copies were quantified by *E1A* gene levels by qPCR.

For subcutaneous tumor rechalle, $1 \times 10^6$ GL261 cells or $1 \times 10^6$ MC38 cells were injected into the right flank of cured mice and age-matched naïve mice.

### Toxicity assessment of AdV[APOA1] in mice and rhesus monkeys

Murine neural stem cells (NE-4C) were seeded in 6-well plates at a density of $5 \times 10^5$ cells/well and cultured with vehicle, ApoA1 (10 µg/ml), or LXR-623 (5 µM) in 1% FBS medium for 3 days. Cell death was determined by Annexin-V/PI.

A long-term toxicity study of intracerebral injection of AdV[APOA1] was conducted at INNOSTAR (Shanghai, China). BALB/c mice were randomly divided into three groups ($n = 64$ mice per group, half male and half female), including the vehicle group, low-dose group ($0.5 \times 10^9$ VP/mouse), and high-dose group ($2.5 \times 10^9$ VP/mouse). Mice were intracranially injected twice (D1, D15) and allowed to recover for 4 weeks to observe the virus-induced toxic reaction, biodistribution, shedding, and changes in antibody titer. Plasma was collected from animals in each group during adaptation period D1 (after grouping), D14, D28, and D44 for anti-AdV[APOA1] antibody titer analysis. Blood was collected on D2, D9, D16, and D44 for IFN-γ, IL-10, IL-2, IL-6, and TNF-α analysis. pUC57-1.8K-hexon-positive plasmid DNA was used as a standard. Rhesus monkeys were randomly divided into four groups (n = 10 monkeys per group, half male and half female), including s.c. vehicle group, s.c. low-dose group, s.c. high-dose group, and i.v. low-dose group. Monkeys were administered once every 2 days for a total of 6 times (D1, D4, D10, D13, and D16) and were dissected on D17 (the end of the administration period) and D46 (the end of the recovery period).

### Quantification and statistical analysis

Statistical significances were determined using the Mann–Whitney test, the paired t test, the unpaired t test, one-way/two-way ANOVA, or the log-rank test by GraphPad Prism 9. The expression differential gene and pathway enrichment analysis of RNA-seq data was performed by R 4.2.1. Sample size was not predetermined using statistical methods. In cases where data was presented without statistics, experiments were conducted at least three times to ensure reproducibility, unless

otherwise specified. All data are presented as mean ± SD. *P* < 0.05 is considered significant.

## Reporting summary

Further information on research design is available in the Nature Portfolio Reporting Summary linked to this article.

## Data availability

The RNA-seq data used in this study have been deposited at the Gene Expression Omnibus under accession codes (SuperSeries GEO accession, GSE201977). Raw LC/MS data for cholesterol-targeted metabolomics are not publicly available due to limitations of LipidALL Technologies (http://www.lipidall.com/). Their rationale is that mass spectrometry files contain content that could harm their commercial interests. If necessary, additional information on raw mass spectrometry in this study (Project: 2021-141-C-04) can be requested from LipidALL by email (sales@lipidall.com, qmchu@lipidall.com, smqi@lipidall.com). The authors declare that the remaining data generated or analyzed during this study are available within the article, Supplementary Information, or Source Data file. Source data are provided with this paper.

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

## Acknowledgements

We thank all lab members and collaborators, especially Prof. Wu Junhua, Dr. Meng Gang, Dr. Xu Tiancheng, Dr. Lin Zhe, FRA. Zhao Siqi, MD. Li Xiaofei, Dr. Yang Fuming, Dr. Wang Shibing, Dr. Chen Anxian, Dr. Wei Min, Dr. Zhang Yenan, Dr. Kang Xiaozhen, Dr. Qian Peng, MD. Li Yuxin, and MD. Xu Chuning for their contributions to this study. We also apologize to other researchers who contributed to this field but whose studies we did not discuss or cite due to limited space. Toxicity Assessment of AdV^APOA1 (NV-A01/NJU-A01) in Mice and Rhesus Monkeys was supported by Nanjing NOVEL Biotech and Nanjing VIROTHER BioPharm. This study was supported by the National Natural Science Foundation of China (82273261, 81773255) to W.J. and the Qiantang Cross Fund (QTJC20220001-B) to W.S.

## Author contributions

The study was conceived and designed by W.S., D.J., and W.J.; Clinical sample collection and biological experiments were performed by W.S., Y.W., and K.L. with assistance from Z.S., W.J., Z.C., H.H., and H.B; Z.S. was responsible for collecting and analyzing the bioinformatics data. The manuscript was written by all authors, with critical revision conducted by W.J. and D.J. All authors gave final approval for submission.

## Competing interests

The authors declare no competing interests.
