## [Peer Review File · Nature Communications]

Oncolytic Viruses Engineered to Enforce Cholesterol Efflux Restore Tumor-Associated Macrophage Phagocytosis and Anti-Tumor Immunity in GlioblastomaEditorial Note: This manuscript has been previously reviewed at another journal that is not operating a transparent peer review scheme. This document only contains reviewer comments and rebuttal letters for versions considered at *Nature Communications*.

REVIEWER COMMENTS

Reviewer #1 (Remarks to the Author):

This is my re-review of Shiqun et al. manuscript on manipulating cholesterol efflux as a strategy to modulate the TME of GBM. I was initially reviewer #1 for the first submission, so you can refer to those for my original comments.

I was initially excited about this publication, contingent on performing some of the experiments and concerns I originally addressed. Overall, I think the authors did a good job addressing my concerns.

I have a comment below I would highly encourage the authors to consider for their future work:

My first comment regarding cell line choices is about the fact that the historically (and most commonly) used cell lines GL-261, CT-2A, and G422 are all mutagen induced tumor cell lines. While I'm aware IDH status is important for tumor stratification, the molecular drivers/suppressors (PTEN, P53, RB, PDGFR, EGFR, CDKs, QKi, etc..) are not well described nor do they match the known drivers of GBM. Contemporary murine GBM modeling involves using inducible models using RCAS-Tva and/or CRE-lox models that remove suppressors/induce drivers. However, I'm also aware these models (and the cell lines derived from them) are not commercially available. I would encourage the authors to contact some of these laboratories in the future for their tumor modeling.

Reviewer #2 (Remarks to the Author):

This is a fascinating and potentially important paper. Some of the additional data have strengthened the findings, but many of the questions, largely around mechanism, have been answered largely by restating what they already did.

Reviewer #3 (Remarks to the Author):

This is a very interesting and potentially impactful study evaluating the efficacy of an oncolytic adenovirus expressing ApoA1. The rationale being that ApoA1 mediated cholesterol efflux would affect TAM and anti tumor immunity is novel to my knowledge. The data look strong and the study is significant.

Major concern is that almost all the data regarding TAMs is generated from mouse models: It is not clear if human GBM derived TAMs also would have similar function. Some key experiments should be repeated in human GBM derived TAM cells.

Sadly the response to prior comments is not satisfactory. For example:

The use of the murine glioma used (carcinogen induced models) models remains a concern.

The need for validation in human TAMs remains unanswered.

The use of OT-1 T cells compared to TAMs is still not addressed.

Other comments are below:

full form for the acronym is missing a lot: for example LXR, LDLR, NBD. Etc.

macrophage phagocytic fragility needs to be defined.

Results section in the sub-heading that "Metabolism Codependency in GBM Facilitates Cholesterol Pooling in the TME" data shows that there is accumulation of cholesterol in tumor IF, that correlates with higher expression of genes related to cholesterol uptake like EGFR and LDL. The metabolic co-dependency that the authors refer to here is not clear??

In general: Details of experiment design should be more clear in results and figure legends

Figure 2e: TEM images are from only one gl261 model. Since gl261 is thought to be immunogenic and a chemo induced model that does not recapitulate GBM biology this should be repeated in human tissue specimens.

The rationale for investigating cholesterol levels in Siglec-10+/PD-1+ 260 TAMs Vs siglec -ve is not explained.

Figure 4A: Missing details and rationale: OT-1T cells are likely derived from spleens of OT-1 transgenic mice, what is the source of TAMs is not clear? Why were splenic T cells compared to TAMs? What tumor model was used to isolate TAMs ??

Almost all the data presented is from mouse model isolated TAMs: it will be important to check if this remains the case for human TAMs.

Fig 5h: In the results section, A brief description of the observed morphological difference in mitochondria between TAMs sorted from GL261APOA1 would be helpful for reader clarity.

Fig 5i: no S.d on the quantification of total protein intensity:

Fig 6: source of BMDMs is not explained.

Point-to-point responses

Reviewer #1 (Remarks to the Author):

This is my re-review of Shiqun et al. manuscript on manipulating cholesterol efflux as a strategy to modulate the TME of GBM. I was initially reviewer #1 for the first submission, so you can refer to those for my original comments.

I was initially excited about this publication, contingent on performing some of the experiments and concerns I originally addressed. Overall, I think the authors did a good job addressing my concerns.

I have a comment below I would highly encourage the authors to consider for their future work:

Point #1

My first comment regarding cell line choices is about the fact that the historically (and most commonly) used cell lines GL-261, CT-2A, and G422 are all mutagen induced tumor cell lines. While I'm aware IDH status is important for tumor stratification, the molecular drivers/suppressors (PTEN, P53, RB, PDGFR, EGFR, CDKs, QKi, etc..) are not well described nor do they match the known drivers of GBM. Contemporary murine GBM modeling involves using inducible models using RCAS-Tva and/or CRE-lox models that remove suppressors/induce drivers. However, I'm also aware these models (and the cell lines derived from them) are not commercially available. I would encourage the authors to contact some of these laboratories in the future for their tumor modeling.

Response:

We appreciate the valuable feedback from Reviewer 1, which would improve the quality of our work. Although our study emphasized TAMs in high cholesterol TME induced by tumorigenesis or treatment, the Reviewer raised valid concerns regarding the genetic heterogeneity of GBM and suggested investigating the influence of driver/suppressor genes on our conclusions. However, creating these strains of mice is time-consuming. Therefore, we validated our findings on TAMs by conducting crucial experiments using fresh tumor tissues obtained from six GBM patients.

We further investigated the efficacy of ApoA1 on human TAMs isolated from GBM tumor tissues from patients who received surgical excision (Department of Neurosurgery, The Second Affiliated Hospital, School of Medicine, Zhejiang University). The patients' clinical information is provided in the revised manuscript as table S2.

First, we analyzed the human GBM derived TAMs by transmission electron microscopy. Similar to our previous findings in murine GBM derived TAMs, we observed distinct heterochromatin clumps in the nuclei and large, poorly cleared phagosomes in the cytoplasm, which indicate the predominance of large or foamy phagosomes containing uncleared organelles or cell debris in human GBM derived TAMs (revised Fig 2e and S2a).

Then we investigated the impacts of cholesterol on phagocytosis of hTAMs. We observed a positive correlation between the expression levels of Siglec-10 and PD-1 and cholesterol contents. The cholesterol levels of Siglec-10+/PD-1+ TAMs were significantly higher than those of Siglec-10-/PD-1- TAMs derived from human GBMs (revised Fig. 2k). These results suggested that elevated cholesterol may hamper phagocytosis of TAMs, which are consistent with the findings in murine GBM derived TAMs (revised Fig 2l).

Finally, we evaluated the impacts of ApoA1 on the phagocytosis of human GBM derived TAMs. Our results showed that ApoA1 treatment enhanced the phagocytosis of human GBM derived TAM (revised Fig 4d), which is consistent with the findings observed in murine GBM derived TAMs.

These results obtained in human GBM derived TAMs, together with our previous findings in murine TAMs, support our claim that ApoA1-mediated cholesterol efflux restored TAM phagocytosis and reactivated TAM-T cell antitumor immunity

~~~~~

**Reviewer #2 (Remarks to the Author):**

**Point #1**

This is a fascinating and potentially important paper. Some of the additional data have strengthened the findings, but many of the questions, largely around mechanism, have been answered largely by restating what they already did.

**Response:**

We sincerely thank the reviewer for the valuable comments. In response, we have further investigated the molecular mechanisms underlying our findings. Specifically, we have demonstrated that 7-KC's inhibition of mitochondrial translation is directly caused by mitochondrial damage (revised Fig 5m and 5n). Our findings revealed that 7-kc treatment caused significant damage to the membrane structure of isolated mitochondria, resulting in content loss and lowered mitochondrial membrane potential. Furthermore, we conducted RNA-seq analysis on BMDMs treated with 7-KC and the mitochondrial translation inhibitor DOX, and our results indicate that 7-KC-mediated inhibition of mitochondrial translation inhibition affects macrophage phagocytosis by inhibiting the TNF signaling pathway, as depicted in the revised Fig 6.

**Other changes in the revised ms:**

1. We have rescaled the layout of Fig 1. The profile image of GBM tissue chip and partial data from Fig S1 were added to revised Fig 1.

2: We further investigated the efficacy of ApoA1 on human TAMs isolated from GBM tumor tissues from patients who received surgical excision (Department of Neurosurgery, The Second Affiliated Hospital, School of Medicine, Zhejiang University). The patients' clinical information is provided in the revised manuscript as table S2.

First, we analyzed the human GBM derived TAMs by transmission electron microscopy. Similar to our previous findings in murine GBM derived TAMs, we observed distinct heterochromatin clumps in the nuclei and large, poorly cleared phagosomes in the cytoplasm, which indicate the predominance of large or foamy phagosomes containing uncleared organelles or cell debris in human GBM derived TAMs (revised Fig 2e and S2a).

Then we investigated the impacts of cholesterol on phagocytosis of hTAMs. We observed a positive correlation between the expression levels of Siglec-10 and PD-1 and cholesterol contents. The cholesterol levels of Siglec-10+/PD-1+ TAMs were significantly higher than those of Siglec-10-/PD-1- TAMs derived from human GBMs (revised Fig. 2k). These results suggested that elevated cholesterol may hamper phagocytosis of TAMs, which are consistent with the findings in murine GBM derived TAMs (revised Fig 2l).

Finally, we evaluated the impacts of ApoA1 on the phagocytosis of human GBM derived TAMs. Our results showed that ApoA1 treatment enhanced the phagocytosis of human GBM derived TAM (revised Fig 4d), which is consistent with the findings observed in murine GBM derived TAMs.

These results obtained in human GBM derived TAMs, together with our previous findings in murine TAMs, support our claim that ApoA1-mediated cholesterol efflux restored TAM phagocytosis and reactivated TAM-T cell antitumor immunity

3. In order to summarize our study more concisely and clearly, we redrew the Graphic Abstract showed in our previous manuscript and integrated it into the revised Fig 8l.

4. Certain parts of the article language have been refined, without altering the conclusion.

5. We generated a new Supplementary Material word file containing the reagent list from the original manuscript along with supplementary data.

~~~~~

Reviewer #3 (Remarks to the Author):

This is a very interesting and potentially impactful study evaluating the efficacy of an oncolytic adenovirus expressing ApoA1. The rationale being that ApoA1 mediated cholesterol efflux would affect TAM and antitumor immunity is novel to my knowledge. The data look strong and the study is significant.

Point #1

Major concern is that almost all the data regarding TAMs is generated from mouse models: It is not clear if human GBM derived TAMs also would have similar function. Some key experiments should be repeated in human GBM derived TAM cells.

Sadly the response to prior comments is not satisfactory. For example: The use of the murine glioma used (carcinogen induced models) models remains a concern. The

need for validation in human TAMs remains unanswered.

Response:

To address the concerns raised by the reviewer, we further investigated the efficacy of ApoA1 on human TAMs isolated from GBM tumor tissues from patients who received surgical excision (Department of Neurosurgery, The Second Affiliated Hospital, School of Medicine, Zhejiang University). The patients' clinical information is provided in the revised manuscript as table S2.

First, we analyzed the human GBM derived TAMs by transmission electron microscopy. Similar to our previous findings in murine GBM derived TAMs, we observed distinct heterochromatin clumps in the nuclei and large, poorly cleared phagosomes in the cytoplasm, which indicate the predominance of large or foamy phagosomes containing uncleared organelles or cell debris in human GBM derived TAMs (revised Fig 2e and S2a).

Then we investigated the impacts of cholesterol on phagocytosis of hTAMs. We observed a positive correlation between the expression levels of Siglec-10 and PD-1 and cholesterol contents. The cholesterol levels of Siglec-10+/PD-1+ TAMs were significantly higher than those of Siglec-10-/PD-1- TAMs derived from human GBMs (revised Fig. 2k). These results suggested that elevated cholesterol may hamper phagocytosis of TAMs, which are consistent with the findings in murine GBM derived TAMs (revised Fig 2l).

Finally, we evaluated the impacts of ApoA1 on the phagocytosis of human GBM derived TAMs. Our results showed that ApoA1 treatment enhanced the phagocytosis of human GBM derived TAM (revised Fig 4d), which is consistent with the findings observed in murine GBM derived TAMs.

These results obtained in human GBM derived TAMs, together with our previous findings in murine TAMs, support our claim that ApoA1-mediated cholesterol efflux restored TAM phagocytosis and reactivated TAM-T cell antitumor immunity

Point #2

The use of OT-1 T cells compared to TAMs is still not addressed.

Response:

The reviewer expressed concern about why OT-I T cells were used in Fig 4a. In revised Fig 3g, we found that APOA1-mediated antitumor immune responses were dependent on macrophages and CD8+ T cells, but not CD4+ T cells or NK cells. Furthermore, ApoA1 reshaped the immune interaction between TAM and CD8+ T cells (revised Fig. 3h), which was performed in both activated Naïve T or OT-I T cells (which exerted specific cytotoxicity on OVA-expressing GL261 GBM cells). To further evaluate how ApoA1 manipulates the TAM-T-cell axis for GBM tumor control, for instance, whether ApoA1 regulated TAM or CD8+ T cell cholesterol levels. Since we have shown that ApoA1 reduced the immunosuppression of TAM on tumor specific OT-1 cells (revised Fig. 3h), we only selected the activated OT-I T cells as representative cells to evaluate the effect of ApoA1 on their cholesterol levels (revised Fig 4a). Our findings suggest ApoA1 primarily facilitates cholesterol efflux from TAMs, indicating a possible ApoA1-TAM-CD8+ T cell axis.

To make it clear, we have added the explanation using OT-1 T cells in the revised ms. We thank the reviewer for this helpful comment.

Other comments are below:

Point #3

Full form for the acronym is missing a lot: for example LXR, LDLR, NBD. Etc.

Response:

We thank the reviewer for pointing out these. As requested, we have added the full form for the acronym in the revised manuscript.

Point #4

Macrophage phagocytic fragility needs to be defined.

Response:

As requested by the reviewer, we further described this concept in lines 174-179, as described below.

Metabolic status controls immune cell differentiation, polarization, mobilization and immunity. For example, intratumoral Treg cells undergo lipid metabolism reprogramming to enforce functional specialization in the TME. Macrophages primarily serve to eliminate foreign bodies, microorganisms, and cellular waste through phagocytosis and digestion. TAMs can exhibit phagocytic fragility, which is marked by the decline of professional phagocytic functions.

Point #5

Results section in the sub-heading that “Metabolism Codependency in GBM Facilitates Cholesterol Pooling in the TME” data shows that there is accumulation of cholesterol in tumor IF, that correlates with higher expression of genes related to cholesterol uptake like EGFR and LDL. The metabolic co-dependency that the authors refer to here is not clear??

Response:

We thank the reviewer for pointing out this. In lines 86-96 of the introduction section, we have briefly described the co-dependent features of cholesterol metabolism in GBM cells, as described below.

Cancer cells can develop a dependence on certain enzymes or transcription factors through altered biochemical and signaling states, even if these factors are not oncogenic. This process, called non-oncogene addiction or co-dependence. Importantly, co-dependencies can be determined by the local biochemical environment in which tumor cells grow. GBM is a highly complex malignancy and remains difficult to treat despite advances in therapy, largely due to inherent barriers associated with the brain as its host organ.

The brain, which contains 20% of the body's cholesterol, relies mainly on astrocyte-driven cholesterol synthesis due to the blood-brain barrier. GBM cells, being metabolically dependent on exogenous cholesterol uptake, are vulnerable to cholesterol modulators that target their cholesterol intake or astrocyte-driven cholesterol efflux.

Point #6

In general: Details of experiment design should be more clear in results and figure legends

Response:

We thank the reviewer for the suggestion. The details of the experimental design are further refined in sections of Materials & Methods, Results, Figure Legends, and Reporting Summary, respectively.

Point #7

Figure 2e: TEM images are from only one gl261 model. Since gl261 is thought to be immunogenic and a chemo induced model that does not recapitulate GBM biology this should be repeated in human tissue specimens.

Response:

As requested by the reviewer, we further examined the TEM images in human GBM derived TAMs. Please see the response to point #1.

Similar to our previous findings in murine GBM derived TAMs, we observed distinct heterochromatin clumps in the nuclei and large, poorly cleared phagosomes in the cytoplasm, which indicate the predominance of large or foamy phagosomes containing uncleared organelles or cell debris in human GBM derived TAMs (revised Fig 2e and S2a).

Point #8

The rationale for investigating cholesterol levels in Siglec-10⁺/PD-1⁺ 260 TAMs Vs siglec -ve is not explained.

Response:

As requested by the reviewer, we added some explanations in the revised manuscript.

In lines 160-201 Result 2 “Cholesterol in the TME Induces Phagocytic Fragility in Monocyte-Derived Macrophages” and in lines 203-237 Result 3 “Cholesterol Levels in TAMs are Positively Correlated with “Don't Eat Me” Receptor Expression”.

In the revised ms we have also explained why we determined the Siglec-10⁺/ Siglec-10⁻; PD-1⁺/PD-1⁻ TAM cholesterol. (1) GBM infiltrative MN-TAMs accumulate cholesterol and become phagocytosis anergy. (2) We found that Siglec-10 and PD-1 expression in TAMs were significantly higher than that in splenic macrophages, whereas SIRP- α expression was lower in TAMs than in peripheral macrophages. (3) We investigated whether cholesterol accumulation increases the expression of Siglec-10 or PD-1, which hampers phagocytosis of TAMs.

Point #9

Figure 4A: Missing details and rationale: OT-1T cells are likely derived from spleens of OT-1 transgenic mice, what is the source of TAMs is not clear? Why were splenic T cells compared to TAMs? What tumor model was used to isolate TAMs ??

Response:

The rationale to use OT-1 cells has been described in the response to the point #2.

We have also described the purpose of our study in the Results section (lines 295-297). The detailed cellular resources has been described in the revised Figure 4 Legend.

Point #10

Almost all the data presented is from mouse model isolated TAMs: it will be important to check if this remains the case for human TAMs.

Response:

As requested, we have checked it in human GBM derived TAMs. Please see the response to the point #1.

Point #11

Fig 5h: In the results section, A brief description of the observed morphological difference in mitochondria between TAMs sorted from GL261APOA1 would be helpful for reader clarity.

Response:

We thank the reviewer for the suggestion. As requested, we have added descriptions regarding the morphology of mitochondria in lines 367-370.

Point #12

Fig 5I: no S.d on the quantification of total protein intensity:

Response:

As requested by the reviewer, we have performed additional 3 independent repetitions of this experiment and, similar results were obtained (revised Fig. 5I). Further quantification analysis showed that cholesterol and 7K-cho overload can reduce the total protein abundance of mitochondria in BMDMs (revised Fig. 5I lower panel).

Point #13

Fig 6: source of BMDMs is not explained.

Response:

We thank the reviewer for reminding this. BMDMs were derived from in vitro differentiation of C57BL/6 mouse bone marrow cells as described in Materials. As requested, we have explained the source of BMDMs in the legend of Figure 6.

Other changes in the revised ms:

1. We have rescaled the layout of Fig 1. The profile image of GBM tissue chip and partial data from Fig S1 have been added to the revised Fig 1.
2. In order to summarize our study more concisely and clearly, we have redraw the Graphic Abstract showed in our previous manuscript and have integrated it into the revised Fig 8l.
3. Certain parts of the article language have been refined, without altering the conclusion.

4. We have generated a new Supplementary Material word file containing the reagent list from the original manuscript along with supplementary data.

REVIEWERS' COMMENTS

Reviewer #3 (Remarks to the Author):

The authors have answered all my queries.

Point-to-point responses (Second time)
REVIEWERS' COMMENTS

Reviewer #3 (Remarks to the Author):

The authors have answered all my queries.

Response:

We appreciate your critical review, which helps to improve the quality of our manuscripts. Finally, thank you for your recognition of our work.